



# European $CH_4$ inversions with ICON-ART coupled to CarbonTracker Data Assimilation Shell

Michael Steiner[1], Wouter Peters[2,3], Ingrid Luijkx[2], Stephan Henne[1], Huilin Chen[3], Samuel Hammer[4], and Dominik Brunner[1]

[1]Empa, Swiss Federal Laboratories for Materials Science and Technology, Dübendorf, Switzerland
[2]Environmental Sciences Group, Dept of Meteorology and Air Quality, Wageningen University and Research, Wageningen, the Netherlands
[3]University of Groningen, Centre for Isotope Research, Groningen, the Netherlands
[4]Institut für Umweltphysik, Heidelberg University, Heidelberg, Germany

**Correspondence:** Michael Steiner (michael.steiner@empa.ch) and Dominik Brunner (dominik.brunner@empa.ch)

**Abstract.**

We present the first application of the atmospheric chemistry and transport model ICON-ART in inverse modelling of greenhouse gas fluxes with an Ensemble Kalman Filter. For this purpose, we extended ICON-ART to efficiently handle gridded emissions, generate an ensemble of perturbed emissions during runtime, and use nudging on selected variables to keep the simulations close to analyzed meteorology. We show that the system can optimize total and source sector specific anthropogenic European $CH_4$ fluxes on a national-scale in an idealized setup using pseudo-observations from a realistic network of measurement stations. We successfully estimated fluxes for agricultural emissions and partially for waste emissions, but were unable to constrain the sum of the remaining emission sources of comparatively low magnitude (mostly fugitives as well as emissions from industry and traffic). Also regions with low emissions and regions with low observational coverage could not be optimized individually for lack of observational constraints. Furthermore, we investigated the sensitivities towards different inversion parameters and design choices with 16 sensitivity runs using the same idealized setup, demonstrating the robustness of the approach when regarding some minimal requirements of the setup (e.g., number of ensemble members). Subsequently, we applied the system to real in-situ observations from 28 European stations for three years, 2008, 2013 and 2018. We used a priori anthropogenic fluxes from the EDGARv6 inventory and a priori natural fluxes from peatlands and mineral soils, inland waters, the ocean, biofuels and biomass burning and geology. Our results for the year 2018 indicate that anthropogenic emissions may be underestimated in EDGARv6 by ca. 25% in the Benelux countries and, to a smaller degree, in northwestern France and southern England. In the rest of the domain, anthropogenic fluxes are corrected downwards by the inversion suggesting an overestimation in the a priori. For most countries, this means that the a posteriori country-total anthropogenic emissions are closer to the values reported to the United Nations Framework Convention on Climate Change (UNFCCC) than the a priori emissions from EDGARv6. Aggregating the a posteriori emissions across the EU27 + UK results in a total of 17.4 $Tg\,yr^{-1}$, while the a priori emissions were 19.9 $Tg\,yr^{-1}$. Our a posteriori is close to the total reported to UNFCCC of 17.8 $Tg\,yr^{-1}$. Natural emissions are reduced from their a priori magnitude almost everywhere, especially over Italy and Romania/Moldova, where a priori geological emissions are high, and over the United Kingdom and Scandinavia where emissions from peatlands



and wetlands were possibly unusually low during the hot and dry summer 2018. Our a posteriori anthropogenic emissions for the EU27 + UK fall within the range estimated by global top-down studies, but are lower than most other regional inversions. However, many of these studies have used observations from different measurement stations or satellite observations. The spatial pattern of the emission increments in our results, especially the increase in the Benelux countries, also agrees well with other regional inversions.

## 1 Introduction

To achieve the long-term goal of the Paris Agreement to limit global temperature increases to well below 2°C, global greenhouse gas (GHG) emissions will have to be reduced drastically in the coming decades. The implementation of the Paris Agreement requires all parties to commit to mitigation measures (described in National Determined Contributions) and to regularly report their anthropogenic GHG emissions, in the form of National Inventory Reports (NIR), to the United Nations Framework Convention on Climate Change (UNFCCC). NIRs are developed from socioeconomic statistics, activity data and emission factors following the guidelines of the Intergovernmental Panel for Climate Change (Eggleston et al., 2006).

Complementary to these bottom-up emission inventories is the "top-down approach", where atmospheric inversions are used for emission estimation from observations (Bergamaschi et al., 2018; Nisbet and Weiss, 2010). Due to the Paris Agreement, the interest in high-resolution inversions with country-scale emission estimates has grown recently. However, top-down emission estimation is still subject to large and poorly quantified uncertainties due to insufficient coverage of measurements, errors in simulated atmospheric transport, representation errors, measurement biases, and other factors. To advance the field, it is therefore paramount to reduce these errors as much as possible and to build modeling systems that properly account for the remaining uncertainties.

Inversions for $CH_4$ have already been made in previous studies. European $CH_4$ emissions have been estimated and compared to bottom-up values in both regional (Bergamaschi et al., 2018, 2022; Petrescu et al., 2023) and global (Deng et al., 2022; Petrescu et al., 2023) inversions, using both surface in-situ measurements and satellite observations. Bergamaschi et al. (2018) compared a total of 7 inversion models with a regional setup for the period 2006 to 2012, all models using harmonized observations. The mean of the estimates of the models for the anthropogenic emissions for EU27 + UK was higher than the reported values for all years, but still within the calculated uncertainty range. Another regional inversion study for Europe was recently presented by Bergamaschi et al. (2022) using a new, nested high-resolution inversion system Flexpart-COSMO TM5 4DVAR. They also compared the results with the Flexpart extended Kalman filter (FLExKF) (Brunner et al., 2012) and with TM5-4DVAR (Meirink et al., 2008) inversions. All three inversion models resulted in higher emissions for 2018 for Germany, France and Benelux than the sum of UNFCCC reported and natural (estimated by GCP) emissions.

European $CH_4$ emissions were also estimated from global inversions operating at lower resolution and often assimilating a smaller set of observations available over Europe compared to the regional systems. Deng et al. (2022) and Petrescu et al. (2023) compared a number of global inversions presented previously by Saunois et al. (2020) with the reported values. While Deng et al. (2022) included all Annex 1 countries (with periodic emission reports) and non-Annex 1 countries (with only



sporadic reports) worldwide, Petrescu et al. (2023) focused on EU27 + UK and also compared regional inversions with the reported values. The results showed that anthropogenic $CH_4$ emissions estimated in regional inversions were generally higher than reported emissions while global inversions were mostly lower. This general tendency was found irrespective of whether

only ground in-situ measurements or satellite observations were assimilated. These results show that there is still little consistency between different inversion results and further work is needed to identify the causes of the discrepancies. Therefore, a new model intercomparison experiment was established by the Atmospheric Tracer Transport Model Intercomparison Project (TransCom), which requires all participating groups to follow a common data protocol ensuring maximum consistency in terms of the usage of observation data, boundary conditions and a priori fluxes. The results presented here for the real data application

are based on simulations following this protocol.

    In inverse modelling, measured atmospheric dry air mole fractions are linked to emissions using an atmospheric transport model (ATM). The most likely set of emissions is determined by minimizing a Bayesian cost function with an inversion algorithm given a prior constraint (usually a bottom-up inventory or flux model) and uncertainties. Different inversion techniques exist (see e.g. Chap. 11 of Brasseur and Jacob, 2017), such as synthesis (Gurney et al., 2002; Baker et al., 2006; Butler et al.,

2010), geostatistical (Michalak et al., 2004; Gourdji et al., 2012), Kalman smoother (Bruhwiler et al., 2005), Ensemble Kalman Filter (EnKF (also includes Ensemble Kalman Smoother), Peters et al., 2005; Tsuruta et al., 2017), and 4-D variational inversion (4D-var, Chevallier et al., 2005; Baker et al., 2010; Bergamaschi et al., 2022). They have been developed to address different trace gases, observations types and spatial and temporal scales. 4D-var and EnKF methods are computationally expensive but have become standard methods today to address large inversion problems.

A limiting factor for analytical synthesis and geostatistical inversions is the dimension of the inversion problem (both the control and observation space), which needs to be sufficiently small to store the related covariance error matrices in computer memory and calculate their algebraic inverse. The 4-D variational approach, on the other hand, where the cost function is minimized by the calculation of its gradient and through an iterative descent, requires an adjoint model, which is often not available for a given ATM. Ensemble Kalman filter and Ensemble Kalman smoother data assimilation (Evensen, 1994, 2003;

Burgers et al., 1998) have the advantages that they can deal with large inversion problems, that no adjoint ATM is required (Kalnay, 2010) and that they return an approximate error covariance matrix. The disadvantage is that the covariance error matrix and the Kalman Gain are only approximated based on a finite ensemble.

    Peters et al. (2005) developed such an ensemble Kalman smoother, which was further implemented in the CarbonTracker Data Assimilation Shell (CTDAS) (van der Laan-Luijkx et al., 2017). It was designed to optimize biospheric and oceanic $CO_2$

fluxes from different biomes and ocean regions on a weekly time-scale by assimilating the $CO_2$ observations from a global network of stations. CTDAS has been applied in subsequent studies to investigate, for example, the carbon budget over North America (Peters et al., 2007), Europe (Peters et al., 2010; Smith et al., 2020), South America (van der Laan-Luijkx et al., 2015) and globally (van der Laan-Luijkx et al., 2017). More recently, CTDAS has been coupled with a Lagrangian particle dispersion model for studying regional carbon budgets (He et al., 2018) and has also been applied to other species like methane (Bruhwiler

et al., 2014; Tsuruta et al., 2019).



A critical requirement for accurate emission estimates by inverse modelling is the quality of the ATM. An attractive new atmospheric model is the global ICOsahedral Nonhydrostatic (ICON) atmospheric modeling framework (Wan et al., 2013; Zängl et al., 2015; Pham et al., 2021), which can be extended with the "Aerosols and Reactive Trace gases"-model (ART), developed at the Karlsruhe Institute of Technology (KIT) (Rieger et al., 2015; Weimer et al., 2017; Schröter et al., 2018) to simulate aerosols and trace gases. ICON-ART can be run from global scale down to cloud resolving scale and has attractive transport properties such as mass conversation, positivity of tracers and the use of recent developments in subgrid-scale transport.

Here we present the first application of ICON-ART in inverse modelling of GHG emissions with CTDAS. For this purpose, we extended ICON-ART with modules for efficient handling of emissions and online (i.e, during runtime) generation of the ensemble of perturbed fluxes, and with a nudging scheme to keep the simulations close to analyzed meteorology. Using an idealized setup with synthetically generated observations, we analyze how well the new, computationally efficient model is suited to constrain European anthropogenic $CH_4$ emissions for individual countries with observations from a European observation network. The sensitivity of the system to different parameters is analyzed in a set of sensitivity experiments. The system is then applied to real observations from a harmonized set of $CH_4$ dry air mole fraction observations from 28 European stations to assess the performance of ICON-ART in terms of atmospheric transport and to demonstrate the capability of the new system to constrain European emissions using this network. A detailed description of our model setup and the methodology is given in Sect. 2. In Sect. 3 we present the results of both applications, with pseudo-observations as well as with real observations. Sect. 4 provides conclusions.

## 2 Model description and methodology

### 2.1 ICON-ART model and simulation setup

#### 2.1.1 Weather and climate model ICON

ICON is a highly versatile non-hydrostatic atmospheric model for global and regional weather and climate simulations developed jointly by the German Weather Service (DWD) and the Max Planck Institute of Meteorology (Wan et al., 2013; Zängl et al., 2015; Pham et al., 2021). It has been used at DWD for operational weather prediction since 2016 and has been coupled with other models including an ocean and a land surface model for climate simulations (Giorgetta et al., 2018). ICON is based on an icosahedral triangular grid, where 20 equilateral triangles of an icosahedron are iteratively split into smaller triangles up to the desired resolution. With such a grid, the problem of singularity at the poles is avoided. To zoom into a specific region, refined grids can be nested into the parent grid, with one additional edge bisection. The model equations are fully compressible and the vertical discretisation is in generalised smooth-level vertical coordinates (SLEVE) (Leuenberger et al., 2010). Tracers in ICON are transported with perfect mass conservation by solving the continuity equation of mass for each tracer consecutively in the vertical with a finite volume method and in the horizontal direction with a simplified flux-form semi-Lagrangian method (Miura, 2007; Lauritzen et al., 2011; Rieger et al., 2015).



### 2.1.2 ART extension for trace gases and aerosols

The ART-model was developed as an extension for ICON at the Karlsruhe Institute of Technology (KIT) (Rieger et al., 2015; Weimer et al., 2017; Schröter et al., 2018) with the aim of simulating aerosols as well as passive and chemically reactive trace gases. The ART module is coupled online with ICON and allows a flexible definition of tracers and processes to be included (Schröter et al., 2018). Since only $CH_4$ was simulated in the present study, all tracers are passive tracers, i.e. they are only transported without radiative feedback on the meteorology and without degradation by the hydroxyl radical (OH). Depletion by reaction with OH is assumed to be negligible given the short residence time of the air masses within the domain of no more than a few days compared to the OH lifetimes of about 10 years.

To simplify and accelerate the treatment of emissions during the simulations, we implemented the Online Emissions Module (OEM) into ICON-ART, which was originally developed for the regional weather and climate model COSMO (Jähn et al., 2020). Unlike the standard offline approach, where numerous input files have to be provided at discrete model time steps, OEM requires only a small number of files at the beginning of a simulation. These files contain annual mean sector-specific 2D emission fields as well as the temporal and vertical profiles for individual emission categories. During the simulation, these profiles are applied online to update the hourly emissions for each species. OEM has recently become an official component of ART (since ART version 2.6.3).

To project the inventory data to the ICON grid, we extended the stand-alone Python package emiproc (also described in Jähn et al., 2020). Emiproc projects emission data of various inventories to the model grid in a mass-conserving manner by calculating the overlap of the source and target grid at every grid cell. It also generates the temporal and vertical scaling profiles.

### 2.1.3 General setup of ICON-ART forward simulations

The ICON-ART simulations were performed in limited-area-mode (LAM) on a grid covering Europe (see Figs. in Sect. 2 and 3). The horizontal grid was R3B06 (see the established grid notation in Zängl et al., 2015), which corresponds to a mean grid spacing $\Delta x$ between neighboring triangles of about 26 km and yields a total of 21,344 grid cells. Vertically, 60 levels were used between the surface and about 23 km altitude. The time step was 120 s. We used ICON in the Numerical Weather Prediction (NWP) configuration with a single-moment microphysics scheme including graupel and the tile approach for soil switched on, considering subgrid-scale land-cover variability with 6 tiles (3 land plus 3 water types).

The meteorological fields were initialised at the beginning of every simulation either with the reanalysis data of ERA5 (for the real data application; Hersbach et al., 2020) or of the ECMWF IFS model (for the pseudo-observation case; https://www.ecmwf.int/en/research/modelling-and-prediction, last access: 11 May 2022). During the simulation, the meteorological fields were weakly nudged in the entire domain towards the 3-hourly reanalysis data to keep the simulated meteorology close to the analyzed meteorology. This required a modification of the ICON code, since ICON in the LAM configuration only allows nudging towards meteorological boundary conditions (density, virtual potential temperature, Exner pressure, specific humidity and wind) near the borders of the domain. Additionally, the LAM-grid was created such that the boundary zone extends over the entire domain. As in the standard scheme, the nudging strength $\alpha_{nudge}$ was set to decrease with the distance of the cell



row $r$ from the lateral boundaries ($r_0$), but the nudging was not restricted to the 8 cell rows closest to the boundaries. Instead it decreased exponentially with an e-folding width of 2 cell rows towards a minimum dimensionless nudging strength of 0.001 applied in the main part of the domain: $\alpha_{nudge} = 0.069\,exp(-\frac{r-r_0}{2}) + 0.001$.

The modeled $CH_4$ dry air mole fractions were constructed with three different types of tracers representing the $CH_4$ background, emissions, and ensemble members. The tracer representing $CH_4$ from emissions ($CH_4^E$) within the model domain was
initialized with a value of zero (or rather with an offset value which is later subtracted in the post-processing) at the simulation start and was updated by OEM with the emissions from the inventories (described in Sect. 2.2.4) at every model time step. The ensemble members used for the optimization scheme in the EnKF are represented as tracers in the model, similar to $CH_4^E$, but with perturbed emissions (see Sect. 2.2). The background $CH_4$ dry air mole fractions ($CH_4^{BG}$) were represented as a separate tracer, which was initialized and updated at the lateral boundaries with data either from the CAMS global greenhouse gas
reanalysis EGG4 with a spatial resolution of about $80\,km$ (for the pseudo-observation case; Inness et al., 2019) or from the CAMS v19r1 inversion product (for the real-data application; available via https://ads.atmosphere.copernicus.eu/, last access: 9 November 2022).

For every 10 d simulation window of CTDAS (see Sect. 2.2.1), one ICON-ART simulation was performed. The model runs were initialized 24 h before each 10 d window to allow for model spin-up. At the end of the 24-hour meteorological spin-up
period, all $CH_4$ tracer mole fractions were overwritten by the initial $CH_4$ conditions produced by the CTDAS system (see Sect. 2.2.1).

### 2.1.4    Adaptations of ICON-ART to CTDAS

To couple ICON-ART with CTDAS in a robust and efficient way, we made a few adaptations to ICON-ART and the simulation setup. CTDAS requires a large ensemble of $CH_4$ tracers to be simulated, each ensemble member corresponding to one specific
perturbation of the state vector (e.g., fluxes and boundary conditions). This is usually achieved by generating an ensemble of input fields, typically one set of perturbed hourly emission maps and boundary conditions per member. For a large ensemble with a few hundred members and hourly emissions, this may results in a very large number or size of input files and correspondingly expensive I/O during the simulation. To overcome this problem, we extended the OEM module with the option to generate an ensemble of perturbed fluxes and corresponding tracers online during the simulation. With this extension, the
only input required at the start of the simulation is the ensemble of perturbed scaling factors ($\lambda$) provided by CTDAS, which greatly simplified and accelerated the simulations. Each scaling factor scales the flux of one emission category (one tracer can experience the emissions of multiple categories) in one region. The regions can be any combination of grid cells (including individual cells) and are defined by a region mask provided as input for OEM (see Sect. 2.2.2). The setups used in this study, e.g. the emission categories and regions, are described in Sect. 2.2.

The generation of the flux ensemble permits negative fluxes, which could result in negative mole fractions of the $CH_4$ tracers. Since none of the available transport schemes in ICON allows for negative tracer mole fractions, we bypassed this problem by adding a constant offset mass mole fraction of $1.2e^{-6}$ to all $CH_4^E$ tracers and subtracting the same offset afterwards from the output.





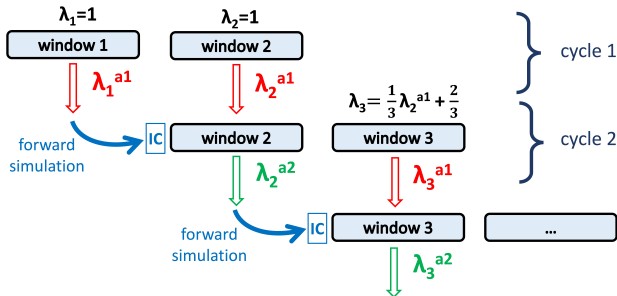

**Figure 1.** Schematic of the ensemble Kalman smoother. The schematic shows the first two assimilation cycles. All remaining cycles are identical to the second one. Each assimilation window has a length of 10 days. One cycle is composed of two windows.

Our implementation of ICON-ART in CTDAS not only allows for the perturbation of fluxes, the ensemble can at the same time also hold perturbed background mole fractions. This allows for the optimization of the background mole fractions, which is described in 2.2.6. Technically, this is achieved by perturbing the $CH_4^{BG}$ tracer mole fractions in different regions of the lateral boundary, with scaling factors provided by CTDAS in a separate file. Similar to the generation of the flux ensemble, a lateral boundary region can be any combination of grid cells (in this case of the lateral boundary cells) defined by a region mask.

## 2.2 CTDAS inversion setup

In this study, we use CTDAS to estimate anthropogenic $CH_4$ emissions either in an idealized setup using synthetically generated atmospheric observations, or in a real data application using quasi-continuous in-situ observations and very few discrete flask samples of $CH_4$ dry air mole fractions. ICON-ART acts as an observation operator, i.e. it connects the surface fluxes to atmospheric $CH_4$ dry air mole fractions. In the following, we describe the setups for the two applications. The description of the idealized setup refers to the reference inversion in which we only optimize emissions. In addition to the reference inversion, we performed sensitivity experiments testing different aspects of the inversion setup as described in Sect. 2.3. These include experiments where we optimized background mole fractions in addition to emissions (see Sect. 2.2.6). Background optimization is also applied in the real data application.

### 2.2.1 Optimization scheme

CTDAS applies an Ensemble Kalman Smoother with a fixed-lag assimilation window (Peters et al., 2005). A schematic of the configuration used here is shown in Fig. 1. Since our model domain is limited to Europe, we assume that the fluxes can affect observed $CH_4$ mole fractions over a maximum of 20 days. Accordingly, CTDAS has been set up to optimize 10-day mean fluxes with a fixed lag of 2, resulting in a total assimilation window length of 20 days. Observations in a given 10-day window can thus constrain the fluxes of the previous and the present 10-day window.



The $s$ scaling factors are optimized using an Ensemble Kalman Smoother as described in Peters et al. (2005), which is based on the Ensemble Square Root Filter presented by Whitaker and Hamill (2002). In the filter, the error covariance matrix **P** (both a priori and a posteriori) of size $[s \times s]$ is represented by information in a smaller dimension $N$, which corresponds to the number of ensemble members, which is set to 192 in our applications. The ensemble is generated from randomly perturbed state vectors with the magnitude and correlation structure of the perturbations, as determined by the a priori error covariance

matrix.

    Since we start with a priori scaling factors with a value of one, the initial perturbation of the scaling factors for the first two windows of the simulation has a mean of $\lambda_{(t=1,2)} = 1$. After assimilation of the observations during these two windows, the first window is simulated again with the optimized scaling factors $\lambda_1^{a1}$ (superscript $a1$ indicates analyzed/optimized once) to provide updated initial $CH_4$ mole fractions for the second cycle. From there, the second cycle uses the optimized scaling factors $\lambda_2^{a1}$

from the first cycle and continues with the third window, which inherits the scaling factors from the previous window with the following state propagation model: $\lambda_{t+1} = 2/3 + \lambda_t^{a1}/3$ (for the pseudo-observation application) or $\lambda_{t+1} = 1/3 + 2\lambda_t^{a1}/3$ (for the real data application). A priori information is thus inserted with a weight of 2/3 (1/3) while a posteriori information from the previous assimilation step is propagated with a weight of 1/3 (2/3) for the pseudo-observation (real-data) application. This weighting is based on the study of van der Laan-Luijkx et al. (2017), where the average of the optimized scaling factors from the

two previous windows were taken into account with a weight of 2/3. Full propagation of a posteriori information is analyzed in a sensitivity inversion. We apply the state propagation model only to the mean scaling factors and not to the individual ensemble members. Each window starts with a new a priori covariance again, hence no information on the uncertainty reduction in the previous windows is taken into account. During the second cycle, observations of the third window are assimilated. The second cycle is completed by simulating the second window again with the now twice optimized scaling factors $\lambda_2^{a2}$, to provide updated

initial mole fractions for the third cycle. The third and all following cycles follow the same principle as the second cycle (Peters et al., 2005, Fig. 1). The main output is thus the sequence of twice-optimized scaling factors (and their uncertainties) for each 10-day window of the simulation period. Error covariances are discussed separately in Section 2.2.5

### 2.2.2   State vector

**Idealized setup**

In our idealized setup, CTDAS separately optimizes the fluxes of three different emission categories, namely agriculture, waste and all remaining anthropogenic emissions including fugitives from industrial plants, gas distribution networks and the energy industries. Natural emissions are not considered in this setup. They are comparatively small accounting for about 15% of the emissions over Europe (Bergamaschi et al., 2018). For every category, the emissions are optimized for 21,344 individual regions, each region corresponding to one grid cell in the domain. The formalism of CTDAS is described in detail in Peters et al.

(2005). As we use a lag of 2, the state vector **x** has 128,064 flux elements in our implementation (2 windows x 3 categories x 21,344 grid cells). Each 64,032 flux elements are scaling factors $\lambda$, which scale the online computed fluxes of the 3 different emission categories in the 21,344 different regions uniformly for all time steps of one 10 day window. Only flux elements are optimized in the reference setup. Simultaneous optimization of fluxes and background mole fraction is analyzed in additional



sensitivity inversions.

### Real data application

In the real data application, only two categories are optimized separately, anthropogenic and natural fluxes. As in the idealized setup, each of the 21,344 grid cells is optimized. Additionally, 8 state vector parameters are used to optimize the background mole fractions (see Fig. 4), resulting in a total state vector size of 85'392 (2 windows x (2 categories x 21,344 grid cells x + 8 background parameters)).

### 2.2.3 Ensemble size

Previous applications of CTDAS used 200, 1500 (Peters et al., 2005), 500 (Tsuruta et al., 2017, 2019) and 150 ensemble members (van der Laan-Luijkx et al., 2017). In Peters et al. (2005) 100-200 model ensemble members were sufficient to represent a state vector of 14,400 parameters, however with substantial spatial correlations to reduce the degrees of freedom. In our setup we use an ensemble size of 192, wherein every ensemble member is represented by one tracer. This number results from the standard maximum number of 200 tracers supported by ICON minus the various water tracers and the background methane tracer. The sensitivity to a smaller size of 50 members was also analyzed. Larger ensemble sizes would require multiple parallel ICON-ART simulations and additional pre- and post-processing steps, which makes the setup more complex and was therefore not tested here. The degrees of freedom (calculated with Eq. 21 in Peters et al., 2005) in our a-priori covariance error matrix was 637 in the idealized setup and 425 in the real data application.

### 2.2.4 A priori fluxes

#### Idealized setup

The anthropogenic $CH_4$ emissions were taken from the European emission inventory TNO_GHGco_v1.1 for the year 2015 at approximately 6 km x 6 km resolution, which is a preliminary version of the CAMS-REG-v4 inventory described in Kuenen et al. (2022). The 15 emission categories were merged into the following three categories: (a) agricultural, (b) waste and (c) all remaining emissions. Figure 2 shows the emissions of the 3 categories as used in our simulations remapped with emiproc to the ICON grid with a locally conservative method. The total emissions within the domain are 11.72 $\text{Tg yr}^{-1}$, 7.16 $\text{Tg yr}^{-1}$ and 6.66 $\text{Tg yr}^{-1}$ for agriculture, waste and the remaining emissions, respectively. Emissions from agriculture are more smoothly and evenly distributed than the other categories. Emissions from the category "remaining" are comparatively large in eastern European countries and a large fraction (28.9%) is emitted from point sources.

#### Real data application

In the application of our system to real observations, we followed the "protocol for the intercomparison of national $CH_4$ emissions estimated by inverse modelling system for Western Europe", an intercomparison effort of the TransCom modelling community. The protocol prescribes the prior fluxes to be used, which were pre-processed and provided on a 0.25° x 0.25° grid. The anthropogenic fluxes (agriculture, waste and fossil fuels) are taken from the EDGAR v6.0 (Crippa et al., 2021) inventory.



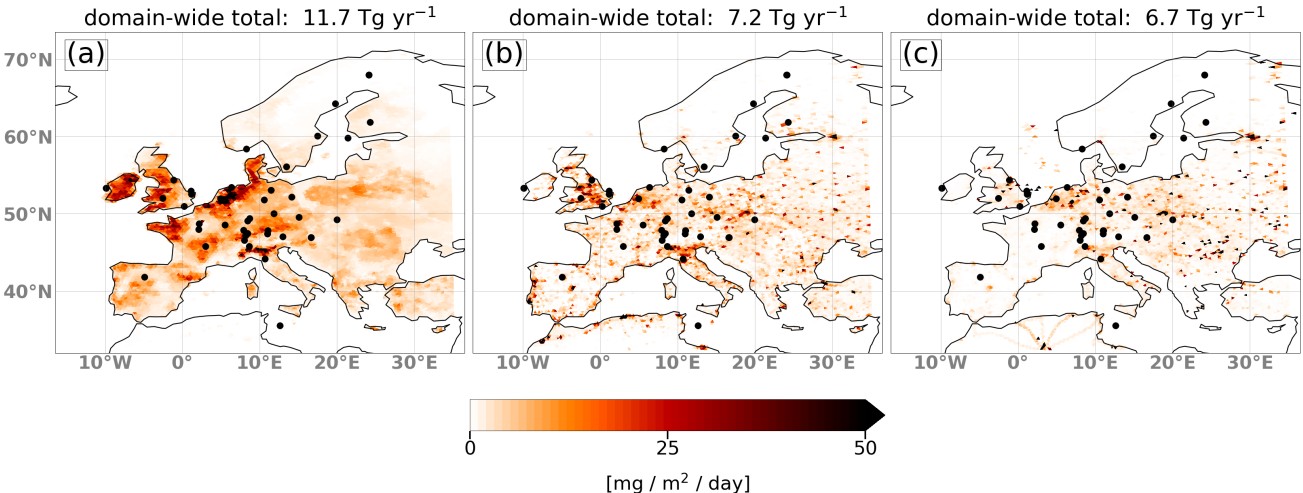

**Figure 2.** Spatial distribution and domain-wide total of $CH_4$ emissions from agriculture (a), waste (b) and remaining sources (c) in the TNO inventory for the year 2018. The black points show the measurement locations used in the pseudo-observation application.

The following natural fluxes are taken into account: peatlands and mineral soils from JSBACH-HIMMELI (Raivonen et al., 2017; Reick et al., 2013) (version 2), inland water (provided by Université Libre de Bruxelles to the GCP-CH4 data set; Saunois et al., 2020), termites (Saunois et al., 2020), ocean (Weber et al., 2019), biofuels and biomass burning (GFED-4.1s; van der
Werf et al., 2017) as well as geological emissions (Etiope et al., 2019) (scaled to a global total of 15 Tg). The pre-processed fluxes with monthly resolution are temporally interpolated to the 10 d windows of our inversion system, such that we have a separate emission file for every 10 d simulation. Since we optimize anthropogenic and natural emissions separately for this real data case (see Sect. 2.2.2), the various natural fluxes were merged to one category. An example of the anthropogenic and natural emissions is shown in Fig. 3 for the period of 11-21 June 2018.
All $CH_4$ emissions are considered to be constant in time over 10-days and are emitted between 0 and 20 m above the surface.

### 2.2.5 A priori error covariance matrix

**Idealized setup**

To reflect the fact that the TNO inventory is scaled to the total emissions reported by the individual countries, we assume that the country-total relative uncertainty does not depend on the size of the country. This results in higher uncertainties at grid
cell level for larger countries. Taking also into account that countries usually report lower uncertainties for emissions from agriculture than for other sectors, we scaled the prior standard deviation of each dimensionless state vector element in a given country such that the relative uncertainty was 50% for country-total agriculture emissions and 80% for waste and remaining emissions. The total prior uncertainty of the emissions in a given region with mask **g**, e.g. a country, can be computed from the



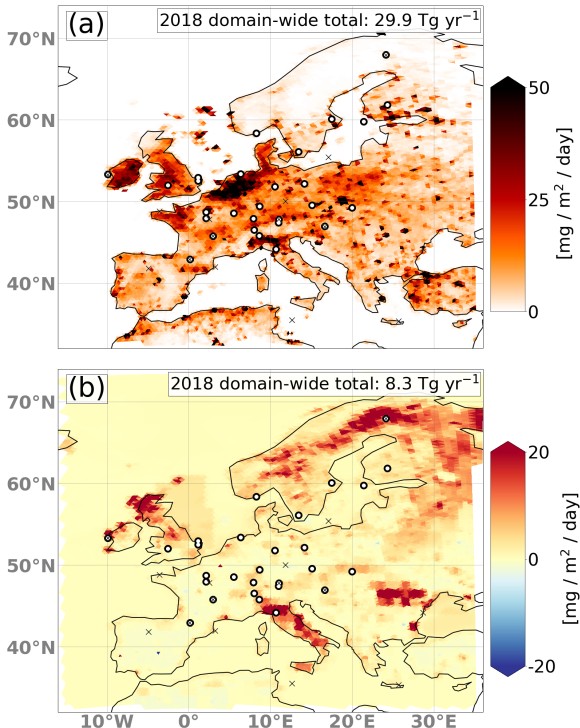

**Figure 3.** Spatial distribution of $CH_4$ emissions from anthropogenic (a) and natural sources (b) used in the real data application for the period of 11-21 June 2018. Overlaid are the measurement locations used in the inversion. The numbers for the domain-wide total fluxes are given for the entire year 2018. The circles show the in-situ measurement locations while the small crosses show the locations of flask sampling.

full prior error covariance matrix $\mathbf{P}^b$ as follows:

$$\sigma^2_{region} = \mathbf{g}^T \mathbf{P}^b \mathbf{g} \tag{1}$$

Other assumptions including uniform (i.e. independent of country) flux uncertainties were tested in the context of the sensitivity simulations (see Sect. 2.3).

**Real data application**

In the real data application, where we use a priori fluxes from the EDGAR inventory for the anthropogenic fluxes and various inventories for natural fluxes, we apply an uncertainty of 100% for each flux in each grid cell. This takes into account that EDGAR applies its own methodologies uniformly to the whole domain and does not scale the total national emissions to the officially reported values.

In both setups, the scaling factors corresponding to the same category but belonging to different regions are correlated de-



pending on the great-circle distance between the centers of the grid cells assuming an exponential decay of the correlation with a length scale $L$=200 km (see Gaspari and Cohn, 1999; Peters et al., 2005), regardless of country borders. Between different categories we assume no correlation, and temporal a-priori correlations between consecutive cycles is also not applied.

### 2.2.6 Background optimization

In the reference inversion of the pseudo-observation application, only emissions but no background mole fractions are optimized. However, if background mole fractions provided at the lateral boundaries from a global model are biased, an inversion system without background-optimization will try to compensate this bias by increasing or decreasing the emissions, which ultimately leads to biased emission estimates.

To address this problem, we implemented the option to optimize background mole fractions alongside the emissions. For 315 this purpose, 8 additional state vector parameters were introduced to optimize the background mole fractions from 8 different inflow regions where $CH_4$ from the global CAMS model enters our model domain (see Fig. 4). We test the capability of this approach to correct for different magnitudes and types of biases in three dedicated sensitivity experiments. In these experiments we introduce different artificial biases with magnitudes of 15 ppb to 90 ppb and set the uncertainty correspondingly. We choose the magnitude of 15 ppb to 90 ppb based on the comparison between simulated $CH_4^{BG}$ driven with the CAMS EGG4 320 product and observations, which reveal substantial biases in this range. Background-optimization is also applied in the real-data application, where we apply an uncertainty of 0.05%, which is roughly 1 ppb $CH_4$. In this case we apply a different state propagation model (see Sect. 2.2.1) for the 8 background parameters than for the emission parameters. We propagate the prior with a weight of 100% to the next window assuming that biases are changing only slowly with time and are therefore similar in subsequent assimilation windows. This allows the system to adjust the background concentration by roughly 1 ppb every 325 10 days. This uncertainty is significantly smaller than what we assumed in the sensitivity simulations. We choose the smaller uncertainty in the real data application as the CAMSv19r1 product used here shows to have almost no biases. In contrast to the CAMS EGG4 product, CAMSv19r1 is the result of a full-fledged inversion system where surface observations have been assimilated to estimate global $CH_4$ fluxes.

### 2.2.7 Localization

To avoid erroneous state vector updates due to spurious covariances between observations and far distant grid cells, the Kalman Gain can be modified using a method called localization. Spatial localization with damping factors decaying exponentially with distance from, or normally distributed around, each observation is possible. An alternative is to only update state vector parameters whose correlation passes a two-tailed t-test, but this proved to be computationally too expensive for our large problem size. To reduce correlations with far distant grid cells, in all our inversions we apply a damping to the Kalman 335 Gain with factors normally distributed around the observation location with a $1\sigma$ standard deviation of 600 km. The effect is illustrated in Fig. 5 for one observation and one emission category. The values of the Kalman Gain matrix can be interpreted as the sensitivity of the observation to upstream emissions from that category.



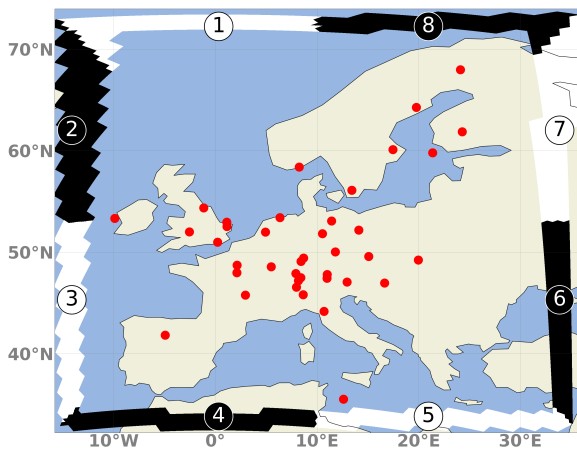

**Figure 4.** Map showing the 8 inflow-regions (labelled 1-8) used for the background optimization.

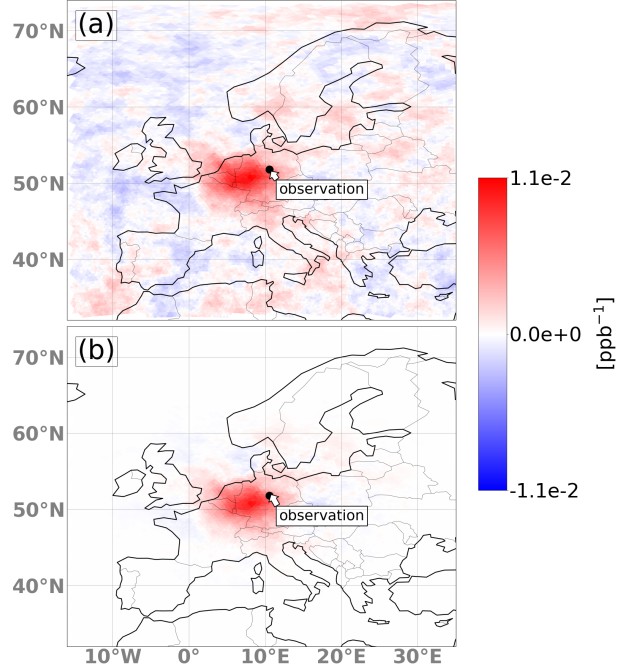

**Figure 5.** The values of the Kalman Gain Matrix for anthropogenic emissions for one single observation at the Torfhaus station in Germany (29 January 2018 12:00 - 16:00) before (a) and after (b) applying localization.



### 2.2.8 Synthetic observations

Pseudo-observations were synthetically generated for the year 2018 by sampling the output of a forward ICON-ART model
simulation at a prescribed set of locations. To make the synthetic experiment comparable to an inversion with real observations, these locations correspond to actual $CH_4$ measurements stations. Only locations with a data-coverage of at least 50% in the year 2018 were selected, which yields a total of 44 stations as shown in Fig. 4, while 5 of them were only assimilated in one sensitivity run. They are listed in Table 1, indicated with the letter "S". As usually done in atmospheric inverse modelling to avoid the difficulties in simulating shallow nocturnal boundary layers, only observations in the afternoon from 11 to 16 UTC
were assimilated. As the model meteorology was identical for the simulations used in the inversions and to produce the pseudo-observations, we did not separate between lowland and mountain stations. In the reference simulation, 5 individual 1-hour averages were assimilated. The impact of using a single 5-hour average afternoon value instead was tested in a sensitivity experiment. In the reference setup, no noise was applied to the synthetic observations. The sensitivity to adding random noise was analyzed in a separate experiment.

The simulation used to produce the synthetic observations followed the reference setup described in Sect. 2.1.3 with one exception. Emissions from the three categories were scaled uniformly for individual large regions in order to create a "true" emission field that is systematically different from the a priori. The performance of the inversion can then be assessed in terms of its ability to reconstruct the true emission field.

  The "true" emissions, i.e. the emissions used in the forward simulation to generate the synthetic observations, correspond
to the remapped TNO emission inventory as described in Sect. 2.2.4, scaled with different scaling factors in 10 regions and separately for the 3 emission categories. These scaling factors are shown in Fig. 6 (left column). The scaling factors were selected randomly from normal distributions with the same variances as used in the a priori covariance matrix (see Sect. 2.2.5) with the additional constraint of only allowing for positive values.

### 2.2.9 Real atmospheric observations

In the real data application, a harmonized set of measurements including quasi-continuous in-situ observations and a few discrete flask samples, is used. The in-situ measurements are from 28 stations while the flask samples are taken at 10 different locations. The flask samples account for ca. 0.5% of the total number of observations. Another 4 in-situ stations are used for validation. Most stations are from the atmosphere network of the Integrated Carbon Observation System (ICOS) (Heiskanen et al., 2022). All 32 in-situ stations and 10 flask sample locations are listed in Table 1, where the assimilated sites are indicated
in the column "R" with "x" and the validation sites with "V". It is important to note that not all time series are complete. In the real data application, we distinguish mountain sites from sites in flat terrain. For the sites in flat terrain, observations in the afternoon from 11 to 16 UTC were assimilated, while for mountain sites the night-time values between 23 and 06 UTC were assimilated.



**Table 1.** List of observation locations used in this study. The column "M" indicates mountain sites while the column "F" indicates if it is a flask sampling location. The columns "S" and "R" indicate if this station is used in the pseudo-observation and real data application. A "V" in the column "R" denotes that is station is used as a validation site.

| ID | station name | M | F | latitude | longitude | elevation [m] | inlet height [m] | network | S | R |
|---|---|---|---|---|---|---|---|---|---|---|
| BEO | Beromünster (CH) | No | No | 47.19 | 8.18 | 797 | 212 | CarboCount CH | x* | |
| BIR | Birkenes (NO) | No | No | 58.39 | 8.25 | 215 | 3 | EBAS | x | x |
| BSD | Bilsdale (UK) | No | No | 54.36 | -1.15 | 380 | 248 | DECC | x* | |
| CBW | Cabauw (NL) | No | No | 51.97 | 4.93 | 0 | 200 | TNO | x* | |
| CIB | Centro de Investigación (ES) | No | Yes | 41.81 | -4.93 | 845 | 5 | NOAA | x | x |
| CMN | Mt Cimone (IT) | Yes | No | 44.17 | 10.68 | 2165 | 7 | WDCGG | x | x |
| GAT | Gartow (DE) | No | No | 53.07 | 11.44 | 69 | 341 | ICOS | x | V |
| HEA | Heathfield (UK) | No | No | 50.98 | 0.23 | 150 | 100 | DECC | x* | |
| HEI | Heidelberg (DE) | No | No | 49.42 | 8.68 | 113 | 30 | INGOS | x | x |
| HPB | Hohenpeissenberg (DE) | No | No | 47.8 | 11.01 | 934 | 131 | ICOS | x | x |
| HPB | Hohenpeissenberg (DE) | No | Yes | 47.8 | 11.02 | 936 | 5 | NOAA | x | x |
| HTM | Hyltemossa (SE) | No | No | 56.1 | 13.42 | 115 | 150 | ICOS | x | x |
| HUN | Hegyhatsal (HUN) | No | No | 46.96 | 16.65 | 248 | 96 | INGOS + HMS | x | x |
| HUN | Hegyhatsal (HUN) | No | Yes | 46.95 | 16.63 | 248 | 96 | NOAA | x | x |
| IPR | Ispra (IT) | No | No | 45.81 | 8.64 | 210 | 16 | ICOS | x | x |
| JFJ | Jungfraujoch (CH) | Yes | No | 46.55 | 7.99 | 3570 | 10 | WDCGG | x | x |
| KAS | Kasprowy Wierch (SVK) | Yes | No | 49.23 | 19.98 | 1987 | 2 | AGH | x | x |
| KIT | Karlsruhe (DE) | No | No | 49.09 | 8.42 | 110 | 200 | ICOS | x | V |
| KRE | Kresin u Pacova (CZE) | No | No | 49.57 | 15.08 | 534 | 250 | ICOS | x | x |
| LAE | Lägern (CH) | No | No | 47.48 | 8.40 | 840 | 32 | CarboCount CH | x* | |
| LIN | Lindenberg (DE) | No | No | 52.17 | 14.12 | 73 | 98 | ICOS | x | x |
| LMP | Lampedusa (IT) | No | Yes | 35.51 | 12.61 | 45 | 5 | NOAA | x | x |
| LUT | Lutjewad (NL) | No | No | 53.40 | 6.35 | 1 | 60 | ICOS | x | x |
| MHD | Mace Head (IRL) | No | No | 53.33 | -9.90 | 5 | 10 | AGAGE | x | x |
| MHD | Mace Head (IRL) | No | Yes | 53.33 | -9.90 | 5 | 21 | NOAA | x | x |
| NOR | Norunda (SE) | No | No | 60.09 | 17.48 | 46 | 100 | ICOS | x | x |
| OPE | Observatoire pérenne (FR) | No | No | 48.56 | 5.50 | 390 | 120 | LSCE | x | x |
| ORL | Orleans (FR) | No | Yes | 47.83 | 2.50 | 170 | 1467-1634 | LSCE | x | x |
| OXK | Ochsenkopf (DE) | Yes | Yes | 50.03 | 11.81 | 1022 | 163 | NOAA | x | x |

* only used in case 13 of the synthetic sensitivity inversions



| ID | station name | M | F | latitude | longitude | elevation [m] | inlet height [m] | network | S | R |
|----|-------------|---|---|----------|-----------|---------------|------------------|---------|---|---|
| PAL | Pallas (FIN) | No | No | 67.97 | 24.12 | 560 | 7 | WDCGG + ICOS | x | x |
| PAL | Pallas (FIN) | No | Yes | 67.97 | 24.12 | 565 | 5 | NOAA | x | x |
| PDM | Pic du Midi (FR) | Yes | No | 42.94 | 0.14 | 2877 | 10 | LSCE | | x |
| PDM | Pic du Midi (FR) | Yes | Yes | 42.94 | 0.14 | 2877 | 0 | LSCE | | x |
| PUY | Puy de Dome (FR) | Yes | No | 45.77 | 2.97 | 1465 | 10 | ICOS | x | x |
| PUY | Puy de Dome (FR) | Yes | Yes | 45.77 | 2.97 | 1465 | 10 | LSCE | | x |
| RGL | Ridge Hill (UK) | No | No | 52.00 | -2.54 | 204 | 90 | DECC | x | x |
| SAC | Saclay (FR) | No | No | 48.72 | 2.14 | 160 | 100 | WDCGG | x | x |
| SMR | Hyytiala (FIN) | No | No | 61.85 | 24.29 | 181 | 125 | ICOS | x | x |
| SNB | Sonnblick (AU) | Yes | No | 47.05 | 12.96 | 3106 | 5 | WDCGG | x | V |
| SSL | Schauinsland (DE) | No | No | 47.90 | 7.92 | 1205 | 6 | WDCGG | x | x |
| SVB | Svartberget (SE) | No | No | 64.26 | 19.78 | 235 | 150 | ICOS | x | V |
| TAC | Tacolneston (UK) | No | No | 52.52 | 1.14 | 56 | 185 | DECC | x | x |
| TOH | Torfhaus (DE) | No | No | 51.81 | 10.54 | 801 | 147 | ICOS | x | x |
| TRN | Trainou (FR) | No | No | 47.96 | 2.11 | 131 | 180 | LSCE | x | x |
| UTO | Uto (FIN) | No | No | 59.78 | 21.37 | 8 | 57 | ICOS | x | x |
| WAO | Weybourne (UK) | No | No | 52.95 | 1.12 | 17 | 10 | UEA | x | x |
| ZSF | Zugspitze (DE) | Yes | No | 47.42 | 10.98 | 2667 | 3 | WDCGG | x | x |

* only used in case 13 of the synthetic sensitivity inversions

### 2.2.10 Model-data mismatch

In the reference configuration of the idealized setup we use a model-data mismatch of 2 ppb + 40% of the anthropogenic signal ($CH_4^A$) in the forward model simulations using the prior emissions for all observations. We assume uncorrelated errors (diagonal observation-error covariance matrix **R**). The impact of a lower uncertainty is assessed in a sensitivity experiment.

For the real data application, we modified the computation of the model-data mismatch for two reasons. Scaling it with $CH_4^A$ or $CH_4^E$ for every measurement generally results in higher uncertainties for measurements with high mole fractions and

hence those measurements that contain the largest regional signal receive the lowest weight in the inversion. More importantly, mismatches between real and simulated pollution events lead to overestimated uncertainties where an event occurred in the model but not in the observations and underestimated uncertainties in the opposite situation. However, we still want to account for the overall higher uncertainty at sites which are more strongly exposed to local emissions. In the real-data application, we therefore apply a model-data mismatch which is station-dependent but constant in time during one year. It is set to 10 ppb +



30% of the yearly mean signal at each station from the sum of the anthropogenic and natural emissions in the forward model simulation with a priori fluxes. We use again a diagonal **R** matrix assuming no correlation between observations.

## 2.3 Sensitivity experiments

Numerous choices such as the length of the assimilation window, the size of the ensemble or the a priori error correlation length scale have to be made before starting an inversion. The settings for the reference inversion were motivated by previous

studies but are to some extent still arbitrary. To evaluate the impact of these choices, additional sensitivity experiments were conducted by systematically varying individual settings. An overview of these experiments is presented in Table 2.

- The setup of the reference inversion described in Sect. 2.2 is labeled as case 1.

- In two further inversions we test the sensitivity towards a lower number of ensemble members (case 2) and a smaller model-data mismatch uncertainty (case 3).

- The influence of adding measurement noise is tested in case 4. We add white noise to the synthetic observations with a variance of 2 ppb+0.4*$CH_4^A$ consistent with assumptions for the **R** matrix.

- The influence of the Kalman Gain localization is investigated by not applying any localization (case 5) or increasing the $\sigma$ from 600 to 1000 km (case 6).

- In case 7, the a priori uncertainties are assumed to be spatially uniform instead of country-specific as in the reference

inversion. An uncertainty per grid cell of 100% is assumed for agriculture and 173% for waste and "rest" emissions, which corresponds roughly to the domain-wide mean uncertainties in the reference case.

- A different state propagation model for the mean state is tested in case 8, which propagates the once optimized state vector of the previous window by 100% instead of 33%.

- The window size is increased from 10 to 20 days in case 9. In case that the emissions affect observed $CH_4$ mole fractions

over more than 20 days, the increase from a 10 d to 20 d window length should improve the results because the number of times the emissions are adjusted towards the a priori emissions is smaller (only every 20 days instead every 10 days).

- Three cases (10-12) are used to test the optimization of the background mole fractions as described in Sect. 2.2.6. We set the relative standard deviation for the 8 dimensionless state vector parameters to 0.02. In case 10, the 8 background tracer mole fractions are uniformly scaled by 0.98. In case 11, instead, 8 different scaling factors are applied to the 8

inflow regions. Case 12 is similar to case 11 but the scaling factors for the 8 regions additionally vary with time. This is probably the most realistic experiment capturing the possibility that large-scale biases in a global model vary both geographically and with time.

- Case 13 tests the sensitivity to the assimilation of 5 more stations, 4 in central Europe and 1 in the UK. These stations (Beromünster (CH), Bilsdale (UK), Cabauw (NL), Heathfield (UK) and Lägern (CH)) are indicated in Table 1 with "x*"

in the column "S".



– In case 14 we assimilated only one afternoon-average value of dry air mole fractions instead of 5 hourly values, which better accounts for the fact that we do not consider temporal correlations in the model-data mismatch.

– In case 15 we use a model-data mismatch which is constant in time but station-dependent. This approach is also used in the real-data application and is motivated in Sect. 2.2.10. In the pseudo-observation case the scaling with each single $CH_4^A$ measurement results in higher uncertainties and hence a low weight in the inversion for measurements with a high emission signal. For the real-data application, it would additionally evoke the problem that mismatches between real and simulated pollution events lead to over-estimations and under-estimations of uncertainties. In this sensitivity inversion, we apply the the same formula for the model-data mismatch ($2\ \mathrm{ppb} + 0.4*CH_4^A$) but replace the instantaneous $CH_4^A$ value with the yearly mean value.

– Finally, case 16 tests the ability of the system to capture temporally varying emissions. For this case, a second set of "true" scaling factors is selected for the emissions in different categories and regions, again randomly drawn from a normal distribution with the same variances as given in the a priori uncertainties. These true emissions linearly move with every 10 d window from the first set (shown in Fig. 6, left column) to this second set such that the second set represents the true emissions at end of June. From July to December, the true emissions linearly move back to the first set of scaling factors.

# 3 Results and discussion

## 3.1 Idealized Setup

### 3.1.1 Reference Inversion

Figure 6 shows the "true" (left) and the yearly mean a posteriori scaling factors (right) for the three optimized emission categories for the year 2018. It can be seen that the optimized state is close to the true state in regions and for categories with high emissions (see Fig. 2), large deviations of $\lambda$ from 1 in the true state and/or good coverage with observation sites, which is the case for agriculture emissions in western Europe and waste emissions in France. The a posteriori scaling factors for the "rest" emissions do not agree well with the true state. It is important to note that the performance of the optimization is strongly depending on the choices of the true state. Especially for the categories "waste" and "rest" with lower emissions than "agriculture", a different set of true scaling factors might yield different results. Also important to mention is the fact that it is impossible for the system in this setup to reproduce the sharp transitions between the 10 perturbed regions, as we prescribe a correlation length of 200 $\mathrm{km}$ that leads to spatial smoothing in the estimated state vector.

Figure 7 shows the total emissions (a) as well as the a priori (b) and a posteriori (c) error of the total emissions summed over all three categories compared to the true state. The improvement (prior - posterior error) in terms of total flux is shown in (d). The results were computed from the mean fluxes for the year 2018. Almost every grid cell with deviations from the true state improves clearly in the a posteriori field, especially in the regions with a high station density in central Europe and on



**Table 2.** Overview of the synthetic simulations. In the column "mdm" the formula used for calculating the model-data mismatch is specified. The column "variation" indicates if the emissions varied temporarily in the simulation that was used to produce the pseudo-observations. The column "bg" indicates if the background mole fractions were varied in the simulation producing the pseudo-observations and optimized in the inversion. Boldface numbers or text signify changes compared to the reference inversion.

| Case | members | mdm | localization | noise | variation | bg | obs/day | window | remark |
|------|---------|-----|--------------|-------|-----------|-----|---------|--------|--------|
| 1 | 192 | 2 ppb + 0.4*$CH_4^A$ | $\sigma$=600 km | None | None | None | 5 | 10 d | reference inversion |
| 2 | **50** | 2 ppb + 0.4*$CH_4^A$ | $\sigma$=600 km | None | None | None | 5 | 10 d | |
| 3 | 192 | **1 ppb + 0.1**\*$CH_4^A$ | $\sigma$=600 km | None | None | None | 5 | 10 d | |
| 4 | 192 | 2 ppb + 0.4*$CH_4^A$ | $\sigma$=600 km | **yes** | None | None | 5 | 10 d | |
| 5 | 192 | 2 ppb + 0.4*$CH_4^A$ | **None** | None | None | None | 5 | 10 d | |
| 6 | 192 | 2 ppb + 0.4*$CH_4^A$ | $\sigma$=**1000 km** | None | None | None | 5 | 10 d | |
| 7 | 192 | 2 ppb + 0.4*$CH_4^A$ | $\sigma$=600 km | None | None | None | 5 | 10 d | **uniform a priori unc.** |
| 8 | 192 | 2 ppb + 0.4*$CH_4^A$ | $\sigma$=600 km | None | None | None | 5 | 10 d | **new forecast model** |
| 9 | 192 | 2 ppb + 0.4*$CH_4^A$ | $\sigma$=600 km | None | None | None | 5 | **20 d** | |
| 10 | 192 | 2 ppb + 0.4*$CH_4^A$ | $\sigma$=600 km | None | None | **yes** | 5 | 10 d | bg uniformly scaled |
| 11 | 192 | 2 ppb + 0.4*$CH_4^A$ | $\sigma$=600 km | None | None | **yes** | 5 | 10 d | 8 $\lambda_{bg}$ |
| 12 | 192 | 2 ppb + 0.4*$CH_4^A$ | $\sigma$=600 km | None | None | **yes** | 5 | 10 d | 8 $\lambda_{bg}$(t) |
| 13 | 192 | 2 ppb + 0.4*$CH_4^A$ | $\sigma$=600 km | None | None | None | 5 | 10 d | **5 more stations** |
| 14 | 192 | 2 ppb + 0.4*$CH_4^A$ | $\sigma$=600 km | None | None | None | 1 | 10 d | **5-hr avg. conc. value** |
| 15 | 192 | **station-dependent** | $\sigma$=600 km | None | None | None | 5 | 10 d | **constant mdm at each station** |
| 16 | 192 | 2 ppb + 0.4*$CH_4^A$ | $\sigma$=600 km | None | **yes** | None | 5 | 10 d | |

the British Isles. The mean absolute error of the yearly mean total fluxes in this setup is reduced by 38.7% in the a posteriori compared to the a priori emissions.

The system is thus capable of optimizing the total anthropogenic emissions of Europe. Also, the system can separately
improve agricultural emissions in Western Europe. However, the system is not or only to a limited degree able to optimize individual $CH_4$ emission categories with comparatively small emissions, such as emissions from waste management or categories with even lower emissions.

### 3.1.2 Sensitivity Simulations

For each of the 16 sensitivity inversions (see Table 2) we quantify the reduction of mean absolute error in the a posteriori
compared to the a priori emissions expressed as a percentage. The results are summarized in Table 3. They show the error reduction for the total emissions based on the 10 d windows $(1 - E_a^e/E_b^e)$ as well as for the 3 emission categories (agri, waste, rest) individually. For the 3 inversions with background optimization, the reduction in the error of the 8 corresponding





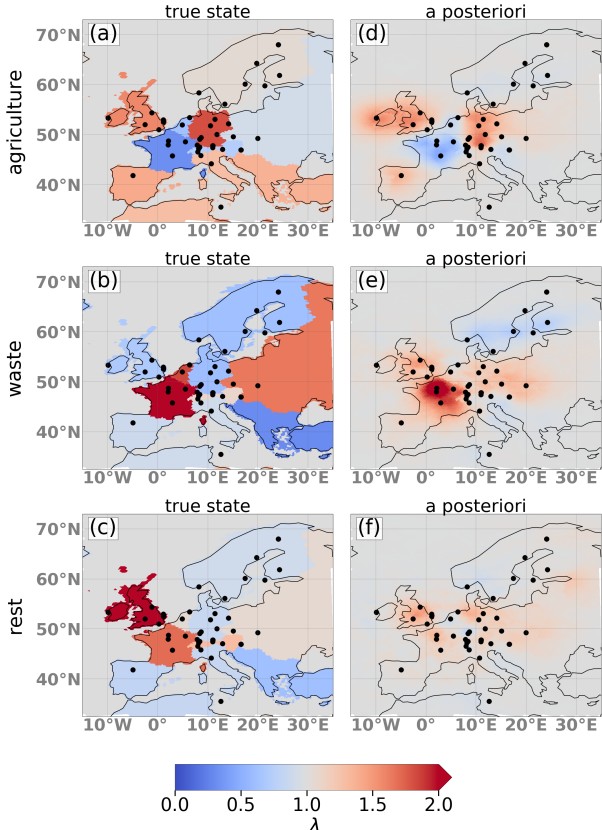

**Figure 6.** Map showing the "true" scaling factors (a-c) and the yearly mean a posteriori scaling factors (d-f) for the three optimized emission categories for the reference inversion.

state vector parameters is shown additionally $\left(1 - E_a^{s,bg}/E_b^{s,bg}\right)$. The reduction of the a posteriori mean absolute error of the mean total emissions fluxes for the considered period (i.e, of the temporally averaged fluxes), is shown in the column

$\left(1 - E_a^{e,mean}/E_b^{e,mean}\right)$. All results are computed for a 7-month period (02.01.2018 - 31.07.2018).

In the reference inversion (case 1), the error in the total a priori fluxes is reduced by 33.3%, while the error of the 7-month mean total fluxes is reduced by 38.7%. The category of agricultural emissions shows by far the strongest error reduction of 37.5%. The a priori flux error of waste emissions is reduced less, by 9.6% while the error for waste emissions is even increased by 11.2%.

The performance for the total fluxes as well as for the individual categories decreases as expected when reducing the number of ensemble members (case 2), adding measurement noise (case 4) or switching off the localization of the Kalman Gain matrix (case 5). Other sensitivity inversions show similar performance as the reference inversion, such as inversions with a longer localization length scale (case 6), uniform a priori uncertainty (case 7), a window length of 20 days instead of 10 days (case



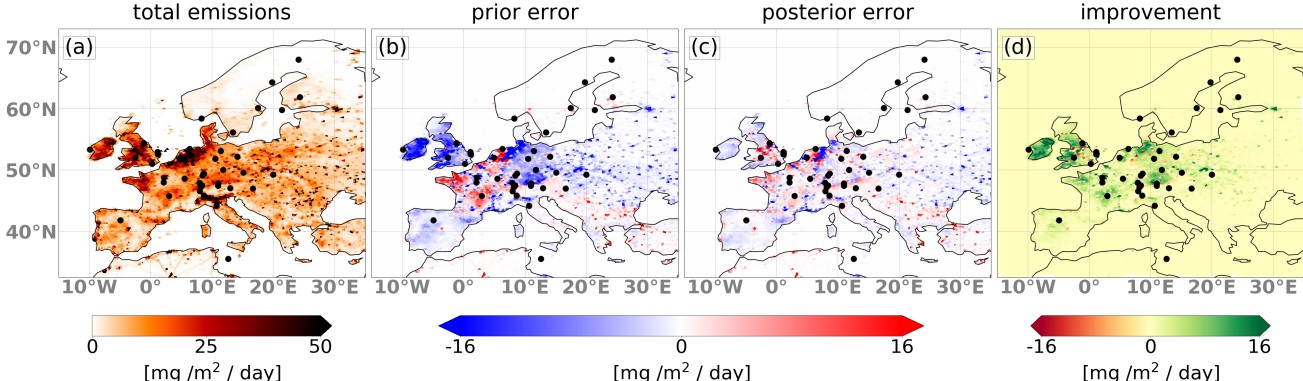

**Figure 7.** Maps showing the total emissions (a), the a priori (b) and a posteriori (c) deviations of the fluxes and the total improvement (d) for the reference inversion.

9), assimilating afternoon average concentrations instead of hourly values (case 14), using a constant model-data mismatch
at each station (case 15), adding 5 more stations in central Europe (case 13) or reducing the model-data mismatch (case 3).
For case 3 this is surprising as a lower model-data mismatch gives more weight to the observations and so should pull the
a posteriori scaling factors closer to the "true" scaling factors. Also for case 13 with 5 more stations this result is not to be
expected. However, the 5 stations are added in central Europe, where the observation network is already dense. Propagating
the state vector by 100% (case 8) improves the error reduction significantly. Such an improvement is to be expected for an
idealized setup with constant emissions, perfect meteorology and without any measurement noise. Disturbing the background
with a constant factor (0.98) and optimizing the background mole fractions with 8 additional state vector parameters (case 10)
influences the optimization of the emissions only minimally. This is due to the fact that the error in the 8 state vector parameters
representing the background mole fractions is reduced by 97.4%, i.e., the error in the background almost vanishes. Using 8
different (but still temporarily constant) scaling factors for the background mole fractions (case 11) reduces the performance of
the emission optimization again very little. The error in the 8 state vector parameters representing the background mole fractions
is now reduced even more strongly by 99.2%. As for the emissions, also the performance of the background optimization is
highly dependent on the chosen "true" scaling factors. Due to the dominating westerly winds, the 2 scaling factors at the
western border of the domain can be constrained the best. Using 8 different scaling factors for the background mole fractions
which additionally change in time (case 12), influences the emission optimization more significantly while the 8 background
state vector parameters are still optimized very well, but apparently projecting some of the concentration variability onto the
background rather than the emissions. In case 16, where the scaling factors of the true state are time-dependent, i.e., where
they gradually move to a second set of scaling factors, the improvement of the total flux is still similar. However, while the
agricultural emissions are optimized less well, the other two emission categories perform much better than in the reference
inversion.



**Table 3.** Statistical results of the sensitivity simulations for the period 02.01.2018 - 31.07.2018. $1\text{-}E_a^e/E_b^e$: the a posteriori mean absolute error of the total emissions fluxes ($E_a^e$) compared to the a priori situation at the beginning of the inversion ($E_b^e$), expressed as a percentage reduction on the prior. The statistics are computed considering every single flux component (every single grid cell and every single assimilation window). $1\text{-}E_a^{e,agri}/E_b^{e,agri}$, $1\text{-}E_a^{e,waste}E_b^{e,waste}$ and $1\text{-}E_a^{e,rest}/E_b^{e,rest}$ show the same as $1\text{-}E_a^e/E_b^e$, but for the 3 catgories individually. $1\text{-}E_a^{s,bg}/E_b^{s,bg}$ denotes the reduction of MAE for the 8 state vector parameters optimizing the background $CH_4$ mole fractions. $1\text{-}E_a^{mean}/E_b^{e,mean}$ finally shows the same as $1\text{-}E_a^e/E_b^e$, but for the mean fluxes during the period instead of considering the individual assimilation windows. Boldface numbers signify the reference inversion.

| Case | Sensitivity | $1\text{-}\frac{E_a^e}{E_b^e}$ | $1\text{-}\frac{E_a^{e,agri}}{E_b^{e,agri}}$ | $1\text{-}\frac{E_a^{e,waste}}{E_b^{e,waste}}$ | $1\text{-}\frac{E_a^{e,rest}}{E_b^{e,rest}}$ | $1\text{-}\frac{E_a^{s,bg}}{E_b^{s,bg}}$ | $1\text{-}\frac{E_a^{e,mean}}{E_b^{e,mean}}$ |
|---|---|---|---|---|---|---|---|
| **1** | **reference** | **33.3%** | **37.5%** | **9.6%** | **-11.2%** | **-** | **38.7%** |
| 2 | 50 member | 17.9% | 30.8% | 1.0% | -42.5% | - | 35.7% |
| 3 | small mdm | 31.8% | 42.5% | 16.5% | -29.8% | - | 48.1% |
| 4 | meas. noise | 7.7% | 26.8% | -0.2% | -71.3% | - | 35.5% |
| 5 | no localization | 16.8% | 30.3% | 0.5% | -71.8% | - | 37.6% |
| 6 | localization w. L=1000 km | 31.0% | 35.7% | 8.3% | -22.5% | - | 39.4% |
| 7 | uniform a priori unc. | 32.5% | 36.8% | 8.7% | -13.8% | - | 38.5% |
| 8 | new forecast model | 43.0% | 44.9% | 22.1% | -9.6% | - | 46.3% |
| 9 | 20 d window | 32.8% | 36.6% | 9.1% | -11.9% | - | 38.0% |
| 10 | bg uniformly scaled | 31.7% | 36.0% | 8.8% | -11.3% | 97.4% | 37.9% |
| 11 | 8 $\lambda_{bg}$ | 30.4% | 34.6% | 8.9% | -13.2% | 99.2% | 36.5% |
| 12 | 8 $\lambda_{bg}$(t) | 18.5% | 28.6% | 3.3% | -41.7% | 96.0% | 35.0% |
| 13 | 5 more stations | 33.7% | 37.7% | 10.5% | -12.0% | - | 39.1% |
| 14 | 5-hr avg. conc. value | 27.9% | 30.2% | 2.9% | -6.5% | - | 30.4% |
| 15 | constant mdm at each station | 32.1% | 36.0% | 8.0% | -8.4% | - | 38.1% |
| 16 | temp. varying emissions | 31.9% | 28.0% | 21.6% | 5.4% | - | 41.2% |

In all cases, the error reduction of the 7-month mean total fluxes is surprisingly similar, meaning that errors in inversions that perform less well when looking at individual assimilation windows (e.g., case 2 with only 50 ensemble members) are averaged out in the mean of a longer period. It can also be seen that for the true state chosen in the cases 1-15, the error in the "rest" emissions always increase, especially for the cases that perform less well in general (e.g., when only using 50 ensemble members, adding measurement noise or switching off localization). This suggests that the emissions in the "rest" category cannot be estimated independently from the others. This is different for case 16, where the true state varies and gradually moves to a second set of scaling factors, which highlights the impact of the chosen true state on the performance of the inversions.




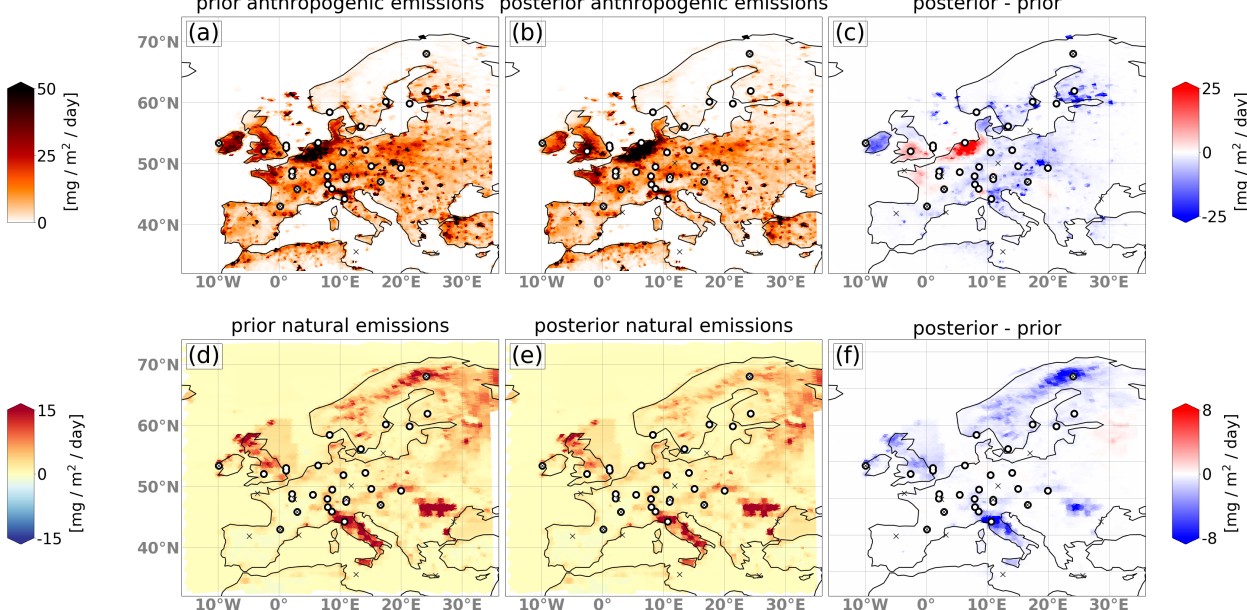

**Figure 8.** Map showing the mean a priori (a, d) and a posteriori (b, e) fluxes for the year 2018 as well as the difference (c, f) for the anthropogenic (a-c) and natural sources (d-f). The results for 2008 and 2013 are shown in the appendix B.

The most important conclusions, that can be drawn from these sensitivity runs for the application with real observations, are that a large number of ensemble members and the localization technique are important for the performance, and that a 10 day window length is a good choice. There are further changes in the real data application setup compared to the reference inversion such as the assumption of a uniform a priori uncertainty and the new model-data mismatch (both explained in Sect. 2.2). The setup in our real data application contains components of the cases 1, 4, 7, 12, 14, 15, and 16.

The sensitivity inversions show what we can expect from the application with real observations: the average total fluxes over a year can be well optimized. The same applies to emission categories with large fluxes (e.g., "anthropogenic emissions"). This is especially true for central and western Europe with good observational coverage. However, the fluxes for individual 10-day windows can be optimized less well (see for example the values for case 4) and categories with small fluxes can hardly be optimized independently. It is also to be expected that the improvement in regions with few measurements (e.g., South-Western and Eastern Europe) will be very small.





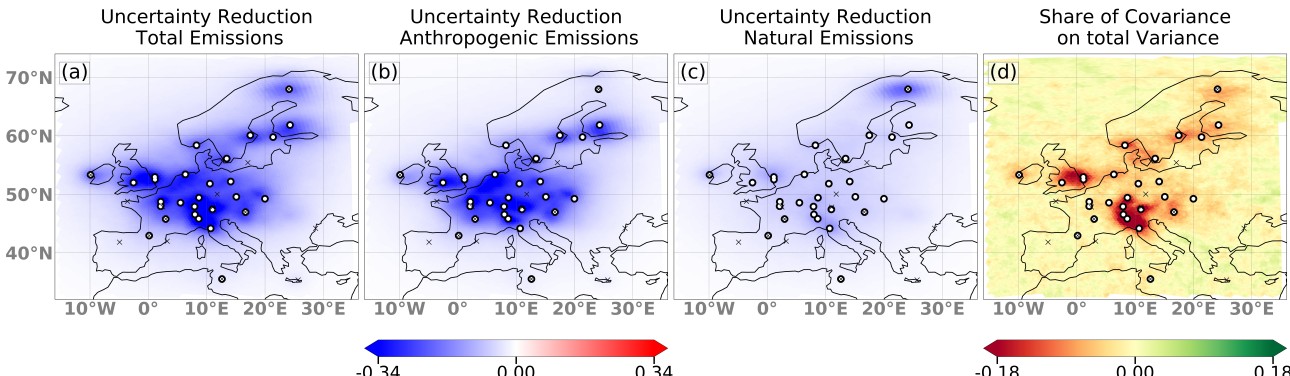

**Figure 9.** Maps showing the differences in the a posteriori uncertainties compared to the a priori uncertainties for the total emissions (a), anthropogenic emissions (b) and natural emissions (c). The fraction of the covariance on the total variance is shown in (d). Optimized fluxes are based on an inversion with real observations.

## 3.2 Real Data Application

### 3.2.1 Posterior Fluxes and Background Mole Fractions.

Figure 8 shows maps of annual mean $CH_4$ emissions for 2018 for the a priori (left), a posteriori (middle) and the difference between the two (right) for the anthropogenic (top) and natural (bottom) fluxes. The anthropogenic emissions show a strong upward correction of up to 25 $\mathrm{mg\,m^{-2}\,day^{-1}}$ in the Benelux countries and a moderate upward correction in northwestern France and southern England. In the rest of the domain, the anthropogenic emissions are corrected downwards, especially over Ireland, but also in the Alpine region and in southern Finland. In the annual mean, the natural fluxes are corrected downward almost everywhere, especially over Italy (up to -22%), the British Isles (-15%), Romania/Moldova (-10%) and Scandinavia (-10%).

Over the Iberian Peninsula, North Africa, and Russia, the increments are generally very low in both categories, as those regions are poorly or not at all constrained by observations. The downward correction of the natural fluxes in Italy and Romania/Moldova is most likely due to the very high a priori geological emissions in these regions. The downward correction of the Scandinavian natural fluxes is possibly due to the unusually hot summer 2018, though the study of Thompson et al. (2022) did not find a clear anomaly of $CH_4$ fluxes in this region in 2018. The a priori wetland and peatland emissions provided by JSBACH-HIMMELI for the year 2018 are a repetition from the year 2017 and hence do not account for the lower $CH_4$ emissions expected for these sources during a hot and dry summer caused by low water table levels (e.g. Bridgham et al., 2013). For the years 2008 and 2013, the system corrects the natural fluxes in Northern Europe much less downward than in 2018. This can be seen in the appendix A in Fig. A1. It shows the time series of the a priori and a posteriori fluxes for the two categories and the three years 2008, 2013 and 2018 for the EU28 countries as well as for northern, western, eastern, and southern Europe.





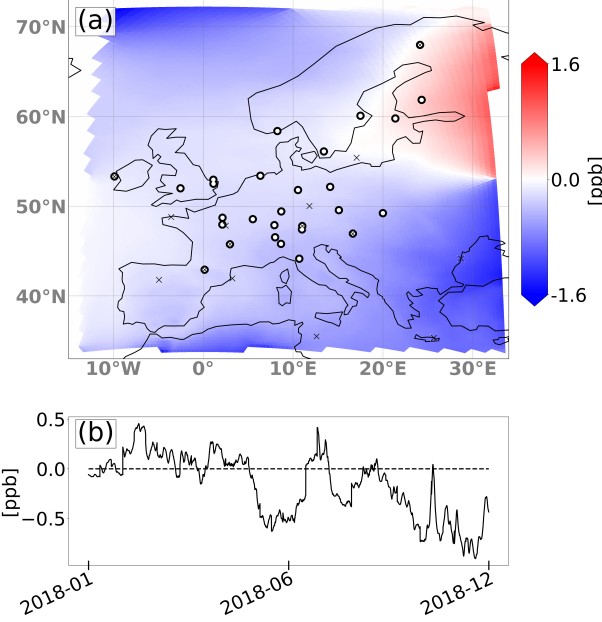

**Figure 10.** The difference of the a posteriori to a priori background dry air mole fractions in the lowest model layer shown as a yearly mean on a map (a) and as a time series of of the domain-wide total (b).

It also shows less downward correction of anthropogenic fluxes in Northern Europe for the years 2008 and 2013 which could be an effect of cross-sector covariances (see Fig. 9d).

Figure 9 presents maps of yearly mean uncertainty reductions (negative values mean a reduction in uncertainty) for the total emissions (a), anthropogenic emissions (b) and natural emissions (c). The fraction of the covariance on the total variance is shown in (d). The total uncertainty is computed from the variances of the two categories as well as the covariances between them, for each gridbox. Since the covariances are typically negative, the total uncertainty reduction is larger than the sum of the uncertainty reduction in the two categories. This is mainly the case in southwestern England, the Alpine region and to a

lesser extent also around the Scandinavian measurement locations. Negative covariances mean that in those regions the system cannot distinguish well between the two fluxes, i.e., an improvement match to observed mole fractions could be achieved by increments of either anthropogenic or natural fluxes in these regions.

As described in Sect. 2.2.6, the system not only optimizes emissions but also background mole fractions in 8 different boundary regions. The effect of this optimization on the mole fractions of the background field in the domain is shown in

Fig. 10, which displays the mean correction of the background field in the lowest model level for 2018, spatially, and as a time series of the domain-wide total. The yearly mean mole fraction difference shows only very small corrections between 0 and 1.6 ppb of the background field, mainly downwards, except for the northeasterly inflow region. The time series shows that the domain-total corrections were slightly upward in spring, while the corrections were downwards during early summer and at



the end of the year. The minor adjustments of the background mole fractions indicate a very small bias in the used reanalysis
product CAMSv19r1, significantly smaller than what we assumed in the sensitivity simulations (cases 10-12).

### 3.2.2 Time Series at Measurement Sites

Figure 11 shows examples of time series for the period 30 March - 13 June 2018 at the stations Lutjewad (NL), Hyytiälä
(FI) and Heidelberg (DE). Simulated dry air mole fractions (from a priori emissions in red and a posteriori emissions in blue)
are compared to observations (in black). Additionally, the assimilated afternoon measurements (black dots) and the simulated
background tracer dry air mole fractions are shown, in dark green for the a posteriori and light green for the a priori. The light
green line is mostly overlayed by the dark green line. At all three stations, the simulation with the a posteriori emissions is
closer to the observations. Furthermore, the a priori background mole fractions match the lowest measured mole fractions in
this period, indicating that the bias of the background field is very low.

Table 4 summarizes the statistics of the model performance at the in-situ measurement stations assimilated in the inversion
(first part of the table). It shows the root-mean squared error ($E$) and Pearson correlation ($r^2$) both for dry air mole fractions at
hours where observations were assimilated (i.e., afternoon values at non-mountain sites and night-time values at mountain sites)
and for all 24 hours per day. For the assimilated stations, the error always decreases and the correlation always increases for the
hours that were assimilated. Also, the error almost always decreases and the correlation mostly increases for the observations
during the entire day. The a posteriori correlation coefficients ($r^2$) for the assimilated observations range from a minimum of
0.59 at the station Ispra (Italy) to 0.93 at the station Ridge Hill (UK). They are above 0.7 at 25 out of 28 stations (89%) and
above 0.8 at 19 stations (68%). These values are similar to those in the high-resolution inversion study of Bergamaschi et al.
(2022) and suggest an excellent model performance typically explaining 70-90% of the observed variance.

The bottom of Table 4 shows the same statistics for the validation sites, which were not assimilated in the inversion. At
these validation sites, the error always decreases and the correlation always increases or stays the same for the afternoon hours
(night-time hours for the mountain-site Sonnenblick (labelled with SNB)). When considering all 24 hourly averages per day,
the error mostly decreases and the correlation mostly increases.

### 3.2.3 National-Scale Emissions

Figure 12 compares 2018 a priori and a posteriori country total emissions for both anthropogenic (left) and natural sources
(right) together with their uncertainties for the 15 largest countries in Europe and the Benelux countries. Anthropogenic emis-
sions are additionally compared to the values reported to UNFCCC for 2018 (year of report 2022). The a posteriori fluxes in the
Benelux countries as well as Germany, France, Norway and Finland are higher than reported. On the other hand, the a posteriori
fluxes are lower in the remaining countries. The strong downward correction of anthropogenic emissions in Italy is likely an
effect of the very high natural (geological) emissions. Although they are corrected downwards strongly in the inversion, the a
posteriori natural emissions in Italy still seem unrealistically high, potentially leading to a misattribution of anthropogenic and
natural emissions in the a posteriori.



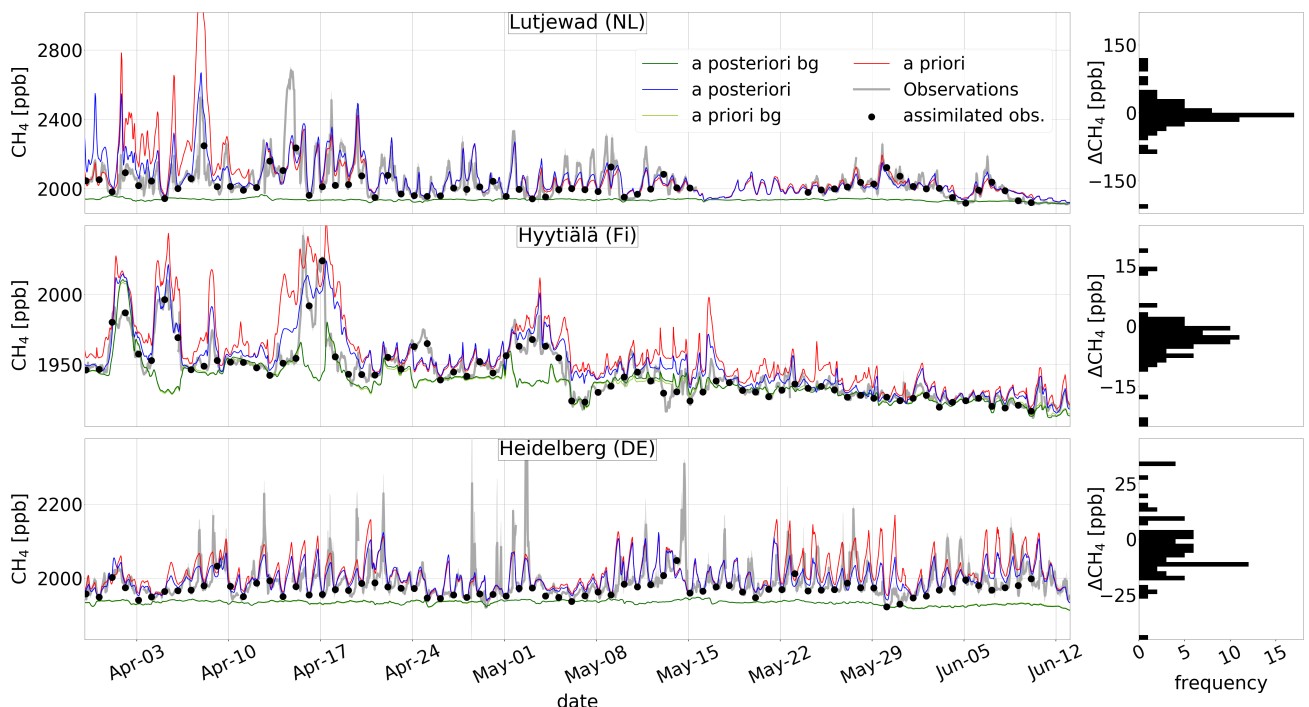

**Figure 11.** Time series of the simulated dry air mole fractions with a priori (red) and a posteriori (blue) emissions. Additionally, the background dry air mole fractions are indicated in dark and light green for the a posteriori and a priori, respectively, as well as the observations (black) with the reported measurement uncertainty (gray shaded, often too small to see) for the period 30 March - 13 June at the stations Lutjewad (NL), Hyytiälä (Fi) and Heidelberg (DE). The assimilated afternoon measurements are shown as black dots. The right column shows the residuals of the assimilated observations in the a posteriori simulation for the same period.

For the sum of all EU27 countries + United Kingdom, the inversion reduces the a priori emissions from $19.9 \, \mathrm{Tg \, yr^{-1}}$ to $17.4 \, \mathrm{tg \, yr^{-1}}$ in the a posteriori. The reported value for the EU27 + UK for the year 2018 is $17.8 \, \mathrm{Tg \, yr^{-1}}$. We performed two additional inversions for the years 2008 and 2013. In these years, the a posteriori fluxes are also lower than the a priori, but still slightly higher than the reported values (2008: a priori: $21.8 \, \mathrm{Tg \, yr^{-1}}$, a posteriori: $21.3 \, \mathrm{Tg \, yr^{-1}}$, reported: $20.5 \, \mathrm{Tg \, yr^{-1}}$; 2013: a priori: $20.5 \, \mathrm{Tg \, yr^{-1}}$, a posteriori: $18.9 \, \mathrm{Tg \, yr^{-1}}$, reported: $18.6 \, \mathrm{Tg \, yr^{-1}}$).

## 4 Conclusions

We developed a new inverse modeling system combining the atmospheric transport model ICON-ART with the ensemble Kalman filter data assimilation system CTDAS, and evaluated its performance in idealized and real $CH_4$ inversions over



**Table 4.** Statistical results at measurement stations assimilated in the inversion and used for validation (bottom). The root-mean squared error ($E$) and Pearson correlation ($r^2$) are given once for dry air mole fractions only at hours where observations were assimilated (denoted with 'assim.') and for all 24 hours per day (denoted with 'all obs.').

| Station | $E_{prior}$ assim. [ppb] | $E_{post}$ assim. [ppb] | $r^2_{prior}$ assim. | $r^2_{post}$ assim. | $E_{prior}$ all obs. [ppb] | $E_{post}$ all obs. [ppb] | $r^2_{prior}$ all obs. | $r^2_{post}$ all obs. |
|---|---|---|---|---|---|---|---|---|
| BIR | 19.2 | 11.9 | 0.92 | 0.95 | 21.3 | 13.9 | 0.89 | 0.91 |
| CMN | 17.1 | 13.6 | 0.78 | 0.80 | 21.2 | 17.6 | 0.69 | 0.73 |
| HEI | 27.4 | 23.0 | 0.85 | 0.87 | 62.0 | 49.7 | 0.66 | 0.68 |
| HPB | 25.3 | 19.3 | 0.80 | 0.89 | 29.9 | 25.8 | 0.75 | 0.81 |
| HTM | 18.5 | 13.6 | 0.93 | 0.93 | 20.3 | 15.7 | 0.91 | 0.92 |
| HUN | 37.2 | 32.3 | 0.66 | 0.73 | 41.3 | 38.9 | 0.59 | 0.65 |
| IPR | 71.2 | 55.4 | 0.59 | 0.75 | 137.2 | 134.8 | 0.48 | 0.44 |
| JFJ | 11.4 | 10.6 | 0.85 | 0.87 | 13.7 | 12.7 | 0.79 | 0.82 |
| KAS | 22.1 | 17.4 | 0.65 | 0.74 | 26.6 | 23.3 | 0.57 | 0.63 |
| KRE | 18.6 | 14.7 | 0.87 | 0.91 | 22.7 | 20.2 | 0.81 | 0.83 |
| LIN | 22.3 | 16.1 | 0.89 | 0.94 | 25.1 | 22.2 | 0.86 | 0.87 |
| LUT | 78.8 | 56.1 | 0.75 | 0.86 | 106.5 | 87.5 | 0.76 | 0.79 |
| MHD | 13.6 | 8.7 | 0.82 | 0.88 | 16.8 | 10.4 | 0.76 | 0.84 |
| NOR | 21.3 | 11.0 | 0.90 | 0.93 | 23.9 | 12.2 | 0.87 | 0.91 |
| OPE | 19.3 | 16.2 | 0.88 | 0.91 | 22.1 | 10.5 | 0.85 | 0.86 |
| PAL | 17.6 | 9.4 | 0.79 | 0.91 | 19.5 | 11.7 | 0.73 | 0.86 |
| PDM | 7.5 | 7.4 | 0.95 | 0.95 | 8.39 | 8.37 | 0.92 | 0.92 |
| PUY | 17.2 | 16.5 | 0.78 | 0.82 | 18.5 | 17.9 | 0.75 | 0.78 |
| RGL | 15.3 | 14.2 | 0.93 | 0.94 | 20.1 | 20.9 | 0.88 | 0.87 |
| SAC | 28.7 | 21.7 | 0.86 | 0.91 | 35.2 | 30.3 | 0.83 | 0.84 |
| SMR | 27.1 | 13.6 | 0.86 | 0.91 | 30.0 | 15.1 | 0.83 | 0.90 |
| SSL | 19.4 | 15.6 | 0.78 | 0.83 | 19.6 | 17.9 | 0.76 | 0.78 |
| TAC | 20.8 | 19.3 | 0.90 | 0.91 | 25.0 | 26.7 | 0.85 | 0.84 |
| TOH | 16.5 | 14.3 | 0.82 | 0.85 | 18.3 | 17.8 | 0.78 | 0.79 |

Europe. For this purpose, we extended ICON-ART with modules for efficient handling of emissions and online generation of the ensemble of perturbed fluxes and with a nudging scheme to keep the simulations close to analyzed meteorology. We showed that with this system, we can optimize total anthropogenic European $CH_4$ fluxes on a national-scale using pseudo-observations in an idealized setup at a realistic network of measurement stations. However, from the three subcategories agriculture, waste





| Station | $E_{prior}$ assim. [ppb] | $E_{post}$ assim. [ppb] | $r^2_{prior}$ assim. | $r^2_{post}$ assim. | $E_{prior}$ all obs. [ppb] | $E_{post}$ all obs. [ppb] | $r^2_{prior}$ all obs. | $r^2_{post}$ all obs. |
|---|---|---|---|---|---|---|---|---|
| TRN | 20.5 | 17.2 | 0.87 | 0.91 | 24.6 | 23.4 | 0.83 | 0.85 |
| UTO | 27.0 | 13.8 | 0.81 | 0.92 | 26.7 | 14.2 | 0.81 | 0.91 |
| WAO | 28.1 | 20.1 | 0.84 | 0.92 | 31.3 | 31.2 | 0.83 | 0.83 |
| ZSF | 15.3 | 12.1 | 0.77 | 0.82 | 16.3 | 13.2 | 0.75 | 0.80 |
| GAT | 16.59 | 16.49 | 0.88 | 0.88 | 21.07 | 20.53 | 0.82 | 0.83 |
| KIT | 25.40 | 20.02 | 0.89 | 0.90 | 30.08 | 27.47 | 0.84 | 0.82 |
| SNB | 15.09 | 12.99 | 0.76 | 0.80 | 16.11 | 13.85 | 0.75 | 0.78 |
| SVB | 17.16 | 11.20 | 0.89 | 0.92 | 17.76 | 11.53 | 0.88 | 0.91 |

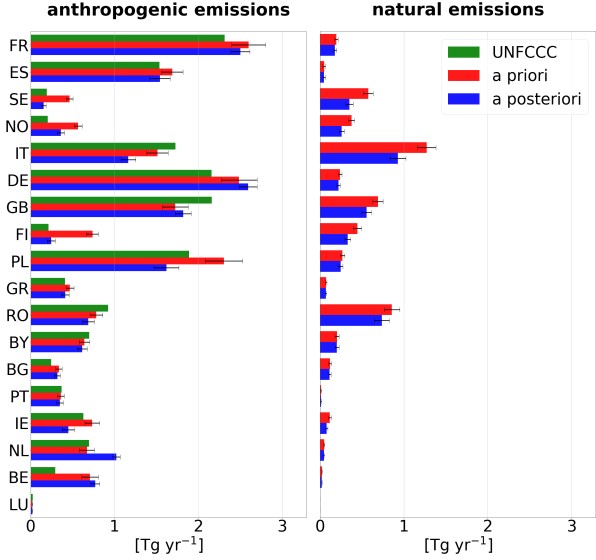

**Figure 12.** Country total emissions for the year 2018 in the a priori (red) and a posteriori (blue) for anthropogenic and natural sources separately as well as the reported anthropogenic emissions to UNFCCC (green, if available). The black bar indicates the total uncertainty as derived from the error covariance matrices. The results for 2008 and 2013 are shown in the appendix B.

and rest, the observations can only successfully constrain the largest source, i.e., agricultural emissions, and partially the second largest source, i.e., waste emissions. The category "rest" (mostly fugitives as well as emissions from industry and traffic) in turn, which has the largest sources in eastern Europe where observation coverage is low, cannot be constrained individually.



Furthermore, we have investigated the sensitivities towards different parameters of the inversions setup with 16 sensitivity runs in the idealized setup. We then applied the system to real in-situ observations from 28 European stations. We used a priori anthropogenic fluxes from the EDGARv6 inventory and a priori natural fluxes from various sources (peatlands, mineral soils, inland water, termites, ocean, biofuels and biomass burning as well as geology).

Our results show that the anthropogenic emissions are significantly underestimated in EDGARv6 for the year 2018 in the Benelux countries (by ca. 25%) and to a weaker extent in northwestern France and southern England. In the rest of the domain, the anthropogenic fluxes are corrected downwards by the inversion. The natural fluxes are corrected downwards almost everywhere, especially over Italy and Romania/Moldova where both regions have very high a priori geological emissions in the data set from (Etiope et al., 2019, scaled to a global total of 15 Tg) as well as in England and Scandinavia (during the

hot and dry summer 2018). For most countries, the a posteriori country-total emissions are closer to the values that were independently determined and reported to UNFCCC than the a priori emissions. The total anthropogenic fluxes for the EU27 + UK is corrected downwards from $19.9\,\mathrm{Tg\,yr^{-1}}$ to $17.4\,\mathrm{Tg\,yr^{-1}}$. The emissions reported to UNFCCC are $17.8\,\mathrm{Tg\,yr^{-1}}$.

The a posteriori anthropogenic emissions in our study are lower than in most other regional inversions (Bergamaschi et al., 2018, 2022; Petrescu et al., 2023). It is important to emphasize that in the various inversions partially other in-situ observations

or satellite measurements were assimilated. While the a posteriori anthropogenic emissions in our study are lower, the pattern of the emission increments are comparable to Bergamaschi et al. (2022). However, our results for anthropogenic emissions are comparable to the results for Europe from global inversions (Deng et al., 2022; Petrescu et al., 2023), both when these assimilate in-situ measurements or satellite observations. It is noticeable that the inversion strongly corrects the overall European emissions downwards such that the independently determined emissions that are reported to the UNFCCC are better

matched. However, the large spread in the various inversions of recent studies shows that there is still substantial uncertainty with inverse emission estimation of European emissions and we cannot trust the absolute values of an individual inversion. The most prominent pattern in our results is the increase over the Benelux countries. This increase of emissions is also visible in other inversion studies, at least in parts of Benelux (e.g. Bergamaschi et al., 2022). This may indicate higher emissions than reported for 2018 in this region due to agricultural emissions.

This overall downward correction is more pronounced for 2018 than for the other two years where we applied our system (2008 and 2013). The reason is most likely the unusually hot and dry summer 2018, which was not taken into account in the a priori peatland emissions and which influences the a posteriori anthropogenic emissions due to cross-sector covariances.

This study presents the new CTDAS-ICON system and shows its application for $CH_4$ in an idealized setup and with in-situ measurements. In future applications, the system could be extended to assimilate additionally satellite observations or

estimate $N_2O$ or $CO_2$ emissions, the latter requiring an extension of ICON-ART with a biosphere-atmosphere exchange flux model. The large uncertainties associated with inverse modelling could be addressed by extending the system with a flow-dependent model-data mismatch, where the emission flux ensemble is coupled to meteorological ensemble for a more realistic representation of the model transport error.



## Appendix A: Further Results 2018

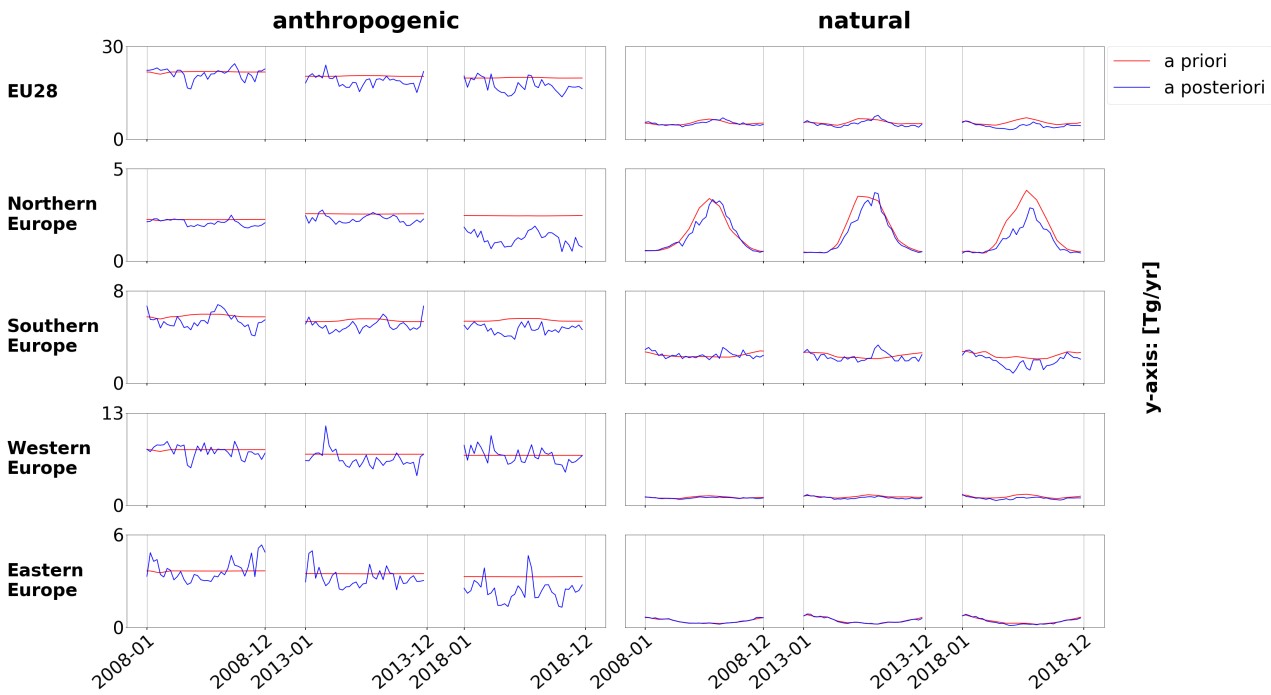

**Figure A1.** Time series for the years 2008, 2013 and 2018 of a priori (red) and a posteriori (blue) CH$_4$ emissions for the EU28 countries as well as for northern (Norway, Sweden, Finland, Denmark, Estonia, Latvia, Lithuania), western (United Kingdom, Ireland, Netherlands, Belgium, Luxembourg, France, Georgia, Switzerland, Austria), eastern (Poland, Czech Republic, Slovakia, Hungary), and southern (Portugal, Spain, Italy, Slovenia, Croatia, Greece, Romania, Bulgaria) Europe.



**Appendix B: Results 2008 and 2013**

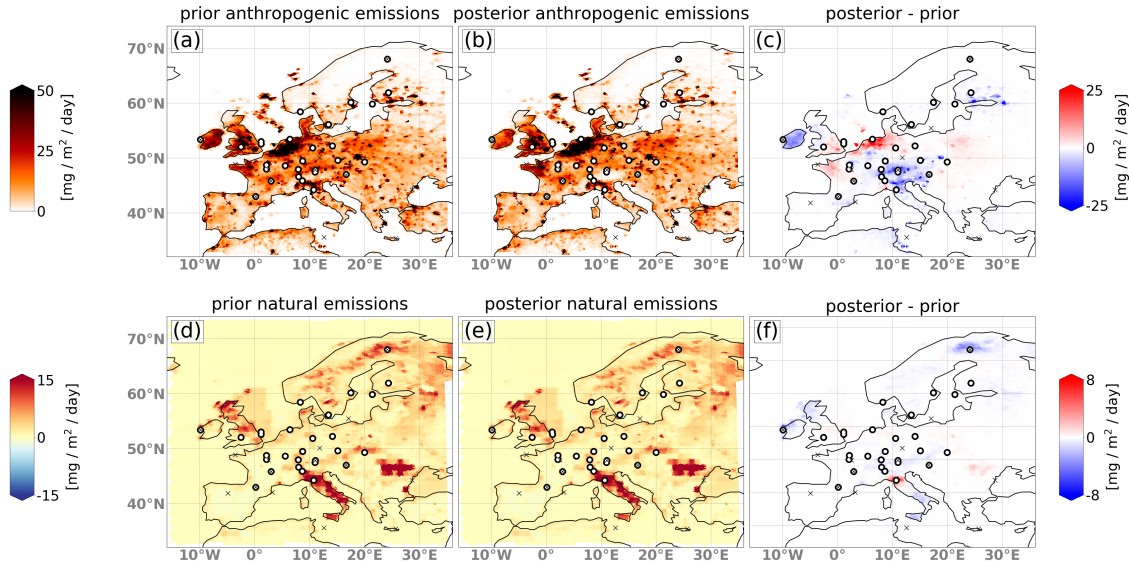

**Figure B1.** As Fig. 8 but for the year 2008.

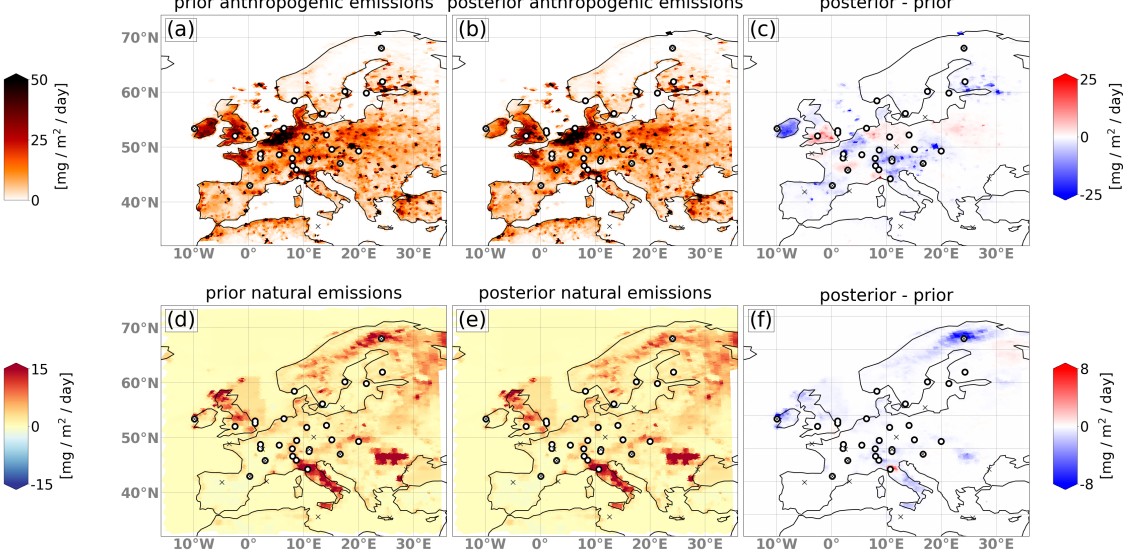

**Figure B2.** As Fig. 8 but for the year 2013.



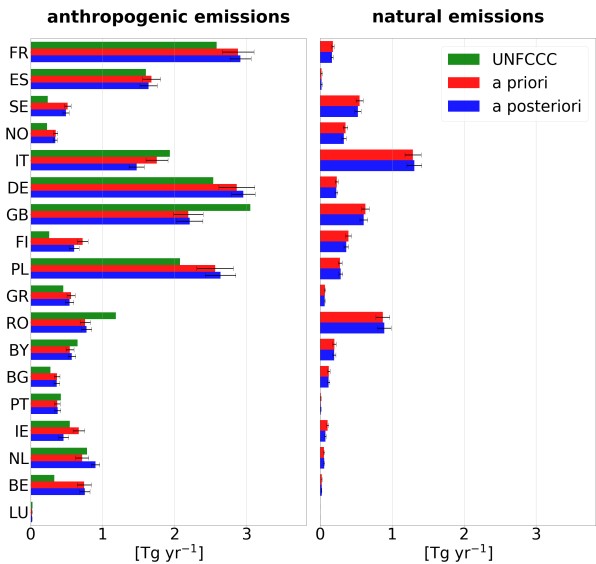

**Figure B3.** As Fig. 12 but for the year 2008.

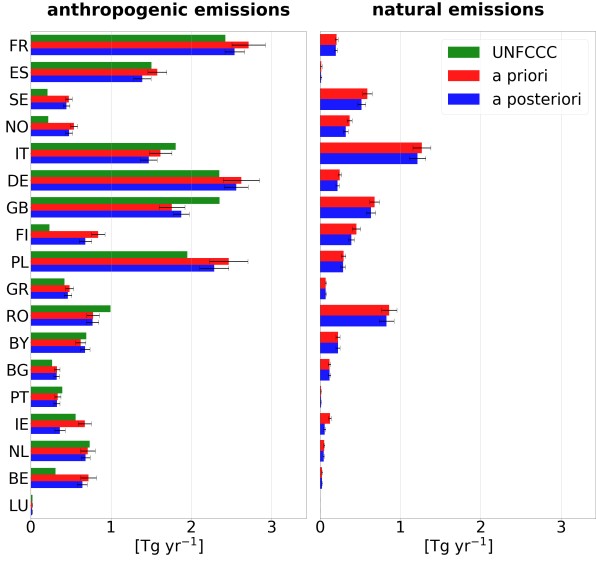

**Figure B4.** As Fig. 12 but for the year 2013.



*Code and data availability.* Access to the git repository containing the official CTDAS code can be obtained by contacting the main developers. The code version used for this study is available in a separate branch of the repository. The ICON code can be obtained from the Max Planck Institute for Meteorology (MPI-M) and the German Weather Service (DWD), who manage the license. For ART, please contact Bernhard Vogel (bernhard.vogel@kit.edu). Our implementations in the ART code are available in the official ART repository. The

atmospheric observations used in the inversions are available at the ICOS carbon portal (https://www.icos-cp.eu), GAW WDCGG data center (https://gaw.kishou.go.jp), NOAA ESRL data (https://www.esrl.noaa.gov/gmd/), and EBAS data center (http://ebas.nilu.no). Information about the pre-processing applied on the observations can be requested from Rona Thompson (rlt@nilu.no).

*Author contributions.* MS and DB initiated the project. MS made the implementations in ICON, coupled ICON with CTDAS, performed and analyzed the simulations and inversion under the supervision of DB. MS, DB and SHe interpreted the results. WP and IL helped with

the coupling of ICON to CTDAS. MS wrote the paper with substantial contribution of DB, SHe, WP and IL. HC and SHa provided the data of the measurement stations Lutjewad and Heidelberg and revised the manuscript.

*Competing interests.* The authors declare that they have no conflict of interest.

*Acknowledgements.* We thank Ivan Mammarella, Petri Keronen and Pasi Kolari for providing the atmospheric observations for the Hyytiälä station. We thank the Environment Agency Austria for providing the atmospheric observations for the Sonnblick station. We thank Cathrine

Lund Myhre, Paolo Cristofanelli, Dagmar Kubistin, Jennifer Müller-Williams, Matthias Lindauer, Tobias Biermann, László Haszpra, Giovanni Manca, Lukas Emmenegger, Lukasz Chmura, Michal Marek, Morgan Lopez, Irene Lehner, Michel Ramonet, Juha Hatakka, Marc Delmotte, François Gheusi, Aurélie Colomb, Simon O'Doherty, Dickon Young, Joe Pitt, Kieran Stanley, Martina Schmidt, Paul Smith, Juha Hatakka, Grant Forster and Cédric Couret for providing the atmospheric observations from the other stations. The Heidelberg measurements were financed as part of the ICOS CRL pilot stations. We thank Lisbeth Florentie and Sander Houweling for the coordination of

the intercomparison exercise within TRANSCOM and setting up the protocol. TRANSCOM received support from the EU-funded CoCO2 project. Furthermore we thank Rona Thompson for the harmonization and pre-processing of the atmospheric observation within the VERIFY project. The Center for Climate Systems Modeling (C2SM) at ETH Zurich is acknowledged for providing technical and scientific support. ICON-ART simulations were performed at the Swiss National Supercomputing Cetre (CSCS) under grants s1152 and s1091.



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
