# Peer review of "European CH4 inversions with ICON-ART coupled to CarbonTracker Data Assimilation Shell"

_EGUsphere, 2023_

## Author Response (AR1)

**Reply to the comment of Referee 1**

We would like to thank the referee for the effort in reviewing our manuscript and for the helpful and constructive feedback. We have taken all comments and suggestions into account and address each of the points raised in this response letter, with reviewer comments in blue and author responses in black. Based on the suggestion for a major revision of referee 2, who particularly raised the point of the discrepancies between the idealized and the real data setup, we did a major revision to our manuscript in which we reconducted the inversions of the idealized case, aligning its setup as closely as possible with the real case. We believe that this harmonization substantially strengthens the manuscript and contributes to a more consistent and concise presentation of our findings.

**GENERAL COMMENT**

The manuscript describes the setup of an inversion system that is used to estimate emissions of methane in Europe based on observations from a surface network. As the presence of methane is one of the major contributors to global atmospheric warming, countries aim at reducing their methane emissions, and inversion systems as presented here are essential to verify whether emission reduction goals are met. The inversion system is well described, and presented results support the conclusions on emission strengths and their uncertainties. These are also in agreement with other inversion studies. Based on this, the manuscript could therefore be published after some minor clarifications.

The inversion system uses the ICON-ART atmospheric transport model, and the CTDAS inversion system that is based on an Ensemble Kalman Filter approach. Experiments with synthetic observations are used to study the sensitivity of the inversion to various configuration choices and parameters. Based on these results, the system is then applied with real observations for 2018 and discussed in detail, while also results for 2008 and 2013 are summarized. This setup of the experiment provides sufficient trust in the inversion systems and the results. The conclusions of the experiments with synthetic data are described around 493-503 as a guideline for the real-data experiment is given, and these are very useful for everybody that considers using an inversion system or analyzing its results. A minor comment is here that is not clear whether the real-data setup has also been fully applied in the synthetic setup; has that been done, or would that not be possible?

As mentioned in the introduction, we repeated all synthetic inversions, choosing the reference setup to be as close as possible to the real setup.

Specifically, we now use the same emissions (anthropogenic emissions from EDGARv6 and various natural emissions as provided by the TRANSCOM intercomparison exercise), assimilate observations at the

same observation sites (and even at the exact same times), optimize the same two categories (anthropogenic and natural) and construct the model-data mismatch in the same way as in the real data application. We believe that this greatly strengthens the link between the two parts of the paper. It also improves the readability of the manuscript as it reduces the text to explaining a single unified setup instead of two.

We further chose a new "true state" of scaling factors that is more consistent with our a priori uncertainty assumptions. Instead of scaling 11 large European regions uniformly, they are now drawn randomly (with a standard deviation of 100%) but correlated in space with a correlation length of 200 km.

In order to save computational costs and to complete this review in a timely manner, we ran the synthetic inversions now only for 2 months instead of 7 months. This is justified as after the 2$^{nd}$ optimization window, the results rapidly approach an equilibrium state with no significant further performance gain in subsequent steps.

Changing the reference setup also required some changes of the sensitivity inversions. Some inversions became obsolete, while other sensitivity-inversions arose logically. The details are presented in the new Table 2 in the manuscript.

The authors take into account uncertainties in emissions throughout the paper. However, in the introduction around line 39, only uncertainties in "top-down" approaches are listed. Could some uncertainties in "bottom-up" approaches be discussed in more detail too? These uncertainties are a main input to the Kalman Filter method used by CTDAS, and section 2.5.5 describes how they are used. Has the configuration been validated using prior simulations with the ensemble to see how much uncertainty in simulated concentrations can be explained by the uncertain emissions? Is the combination of uncertain emissions and representation errors sufficient to explain all observation-vs-simulations differences?

We agree that bottom-up uncertainties are a key input to the Kalman Filter as they are the basis for the a priori error covariance matrix. Unfortunately, no spatially resolved uncertainties are provided with bottom-up inventories. Globally, Solazzo et al. 2021 estimated the uncertainties in EDGAR CH4 emissions to be between −17% to +23% at 1σ, but with a great variability across individual sectors.

In contrast to global or country-wide annual uncertainties, we use uncertainties per grid cell and for each 10 day window, which are extremely difficult to derive from the studies mentioned above. Super et al. (2020) and Choulga et al. (2021) laboriously calculated uncertainties for the a priori covariance matrix for CO2. However, it is difficult to derive uncertainties or correlation lengths for our covariance matrix. Doing this in this detail for our study would go beyond the scope of this work. However, both studies show that the relative uncertainties increase when they are broken down into individual grid cells and shorter time periods. Therefore, our assumption of 100% per grid cell is certainly a simplified assumption, but we believe it is reasonable.

We have supplemented the text in the introduction with a few sentences about bottom-up uncertainties. We have also supplemented Figure 11, which shows the country total emissions, with the uncertainties reported by the countries to the UNFCCC. This allows us to compare the reported uncertainties with our a priori (and a posteriori) uncertainties at the country level. In this comparison, it is evident that the country-level uncertainties calculated from our covariance matrix are reasonably comparable to the reported uncertainties, with some reported uncertainties even being substantially larger (e.g., in France).

One way to evaluate whether the chosen a priori uncertainty and the chosen mdm are reasonable is the chi-square test, which compares the assumed uncertainties with the actual differences between observations and simulation (Michalak et al., 2005). An innovation chi-squared ( $(\mathbf{y}\text{-}H(\mathbf{x}))^2 / (H\mathbf{P}^b H^T + \mathbf{R})$ ) compares the a priori model performance with the total covariance (so assumed flux uncertainty and model-data mismatch) and a value of 1 indicates consistency. The innovation chi-squared for the 3 years in our simulations with real observations are:

| 2008 | 1.13 |
|------|------|
| 2013 | 0.77 |
| 2018 | 0.79 |

Table 1

The innovation chi squared for 2008 is slightly above 1 (indicating a slight underestimation of the total covariance) while it is slightly below 1 for 2013 and 2018. Overall, the values are close to 1, indicating that we have chosen reasonable uncertainties and model-data mismatches in our inversions.

Table 2 additionally shows the regular chi squared ( $(\mathbf{y}\text{-}H(\mathbf{x}))^2 / \mathbf{R}$ ). This value should get closer to 1 in the posterior compared to the prior.

|      | a priori | a posteriori |
|------|----------|--------------|
| 2008 | 2.24     | 1.71         |
| 2013 | 1.64     | 1.08         |
| 2018 | 1.60     | 0.97         |

Table 2

For the years 2013 and 2018 the a posteriori statistics are very close to 1. The value of 1.71 for 2008 indicates that we underestimated the model-data mismatch for that year, which could be the result of poorer model performance at that time. Alternatively, the flux prior could have been closer to the true value than we assumed in the posterior covariance matrix.

Total emissions are compared to results from other inversion studies, and show similar directions of changes in a priori emissions as other inversion studies. For example, this and other studies suggest that emissions in BeNeLux countries are actually higher than currently estimated in the inventories. But to what extent are these results influenced by emissions close to the observation sites? For example for BeNeLux, the estimated emissions are probably strongly constrained by the observations from Cabauw and Lutjewad; could it be that these sites are under heavy influence of nearby sources, and therefore not representative for a larger area?

As specified in the protocol of the TRANSCOM intercomparison exercise, we use data of the Lutjewad station but not of Cabauw. Cabauw was used in our idealized studies, but as we now made this setup similar to the real data setup, we are not using the Cabauw station at all anymore.

The inlet height of 60 meters at Lutjewad (in Cabauw it would be 207 m) should ensure a fairly large footprint and should therefore represent the more distant emissions and not the local emission sources. While the Lutjewad station is surrounded by agricultural lands, it is far from factories and hence from industrial sources. But we are convinced that local agricultural emissions are not measured either at 60 meters inlet height.

A comparison with the modeled concentrations at this station can also provide an indication in this regard. If the station were "contaminated" by local emissions, the comparison with the simulation would be poor. As Table 4 of the manuscript shows, the mean a priori error for Lutjewad is 78.8 ppb while the a priori Pearson correlation $r^2$ is 0.76. This means that the error is certainly slightly higher than at most other stations, but this is to be expected as the concentrations are higher overall due to the high emissions in the Benelux region. The correlation in turn is well comparable with other stations.

Furthermore, Menoud et al. (2020), who used methane measurements from the Lutjewad station to determine sources of methane emissions around Lutjewad, found that the two models CHIMERE and FLEXPART-COSMO were able to simulate the variations of CH4 concentrations fairly well, which largely excludes local influences.

The results compare the posterior emissions with the a priori estimates (EDGAR) and with UNFCCC reporting. Although not really within the scope of the manuscript, could the authors say something about the origin of the differences between these two?

In contrast to the TNO inventory, the country-total emissions in the EDGAR inventory are not scaled to the reported emissions. Differences may arise as different, independent methodologies to estimate emissions are used in EDGAR. EDGAR, for example, largely relies on (Tier-1) IPCC default emission factors (Solazzo et al., 2021), whereas the countries may use country-specific factors (Tier-2 or Tier-3). Furthermore, national inventory agencies often have access to more detailed activity data (e.g. on agricultural activity or the waste sector) for their own country than available to EDGAR.

Lines 153-154: What is meant with "boundary zone extends over the entire domain"? That the interface where nudging is applied is over all domain?

Yes, that is correct.

Line 185: The formulation allows negative fluxes in emissions. Is this an artefact of the method, or meant to allow soil sinks? Couldn't a log-normal emission perturbation be used in the Ensemble Kalman Filter approach, which in principle allows non-linear state-space models?

When an ensemble is generated with a certain distribution, negative scaling factors, and hence negative fluxes, are possible. We don't cut those fluxes at 0, because this would lead to statistical biases. Also, the natural fluxes can indeed be negative.

A log-normal emission perturbation would be possible, and there exist inversion systems that use such a log-normal distribution (e.g. Brunner et al., 2017).

Figure 9 shows a map "fraction of covariance on total variance", which is explained in lines 525-532. This is a well chosen way to illustrate where the inversion system might be able to distinguish between anthropogenic and natural emissions. But isn't this also related to how dominant each of these sectors are at a certain location? Would a map of the ratio between natural and anthropogenic emissions show a similar pattern?

The map of the ratio anthropogenic/natural emissions does not show the same pattern as the map in Fig. 9d. The decisive factors for a large a posteriori covariance between the two categories are that both categories have significant emission fluxes and that the region is well constrained or in other words, that we also have a significant error reduction, i.e. there are enough observations in the vicinity. The greater the error reduction in both categories at the same time, the stronger the correlations between the categories that were zero a priori become.

If we compare the map in Fig. 9d with the distribution of the two emission categories (with the observation stations) in Fig. 8, we see that this is exactly the case over northern Italy.

Figure 10 suggest that May 22- June 12 is period where site Hyytiala receives background concentrations only, as the gray line with hourly observations is nearly flat. The simulated concentrations show still variation however. Does this mean that the model assumes presence of local sources that are actually not present? Or is this due to limited grid cell sizes?

Indeed, in the period May 22 – June 12, the observations in Hyytiala are nearly flat and much lower than the a priori simulated concentrations. It is certainly a period that is dominated by the "background signal". However, the observations are not quite as flat as the modelled background concentrations. This means that most likely the station still sees some signal of emission fluxes on top of the background concentration, but indeed the a priori emissions around the station seem to be overestimated. The time series shows that the inversion quite successfully corrects for this as the a posteriori concentrations are substantially lower and closer to the observations.

**STYLE AND GRAMMAR**

The paper is very well written and easily readable. Only some very minor remarks here.

- Line 129: Shouldn't this be "CH4 lifetimes of about 10 years"?

That is correct. We fixed this.

- Shouldn't "a priori" and "a posteriori" be in Italic font?

In the Copernicus "English guidelines and house standards" it is stated that common Latin phrases such as "a priori" are not italicized.

**References:**

Brunner, D., Arnold, T., Henne, S., Manning, A., Thompson, R. L., Maione, M., O'Doherty, S., and Reimann, S.: Comparison of four inverse modelling systems applied to the estimation of HFC-125, HFC-134a, and SF$_6$ emissions over Europe, Atmos. Chem. Phys., 17, 10651–10674, https://doi.org/10.5194/acp-17-10651-2017, 2017.

Choulga, M., Janssens-Maenhout, G., Super, I., Solazzo, E., Agusti-Panareda, A., Balsamo, G., Bousse-rez, N., Crippa, M., Denier van der Gon, H., Engelen, R., Guizzardi, D., Kuenen, J., McNorton, J., Oreg-gioni, G., and Visschedijk, A.: Global anthropogenic $CO_2$ emissions and uncertainties as a prior for Earth system modelling and data assimilation, Earth Syst. Sci. Data, 13, 5311–5335, https://doi.org/10.5194/essd-13-5311-2021, 2021.

Menoud, M., van der Veen, C., Scheeren, B., Chen, H., Szénási, B., Morales, R.P., Pison, I., Bousquet, P., Brunner, D. and Röckmann, T., 2020. Characterisation of methane sources in Lutjewad, The Netherlands, using quasi-continuous isotopic composition measurements. *Tellus B: Chemical and Physical Meteorology*, 72(1), p.1823733.DOI: https://doi.org/10.1080/16000889.2020.1823733

Michalak, A. M., Hirsch, A., Bruhwiler, L., Gurney, K. R., Peters, W., and Tans, P. P. (2005), Maximum like-lihood estimation of covariance parameters for Bayesian atmospheric trace gas surface flux inversions, *J. Geophys. Res.*, 110, D24107, doi:10.1029/2005JD005970.

Solazzo, E., Crippa, M., Guizzardi, D., Muntean, M., Choulga, M., and Janssens-Maenhout, G.: Uncertain-ties in the Emissions Database for Global Atmospheric Research (EDGAR) emission inventory of green-house gases, Atmos. Chem. Phys., 21, 5655–5683, https://doi.org/10.5194/acp-21-5655-2021, 2021.

Super, I., Dellaert, S. N. C., Visschedijk, A. J. H., and Denier van der Gon, H. A. C.: Uncertainty analysis of a European high-resolution emission inventory of $CO_2$ and CO to support inverse modelling and network design, Atmos. Chem. Phys., 20, 1795–1816, https://doi.org/10.5194/acp-20-1795-2020, 2020.

**Reply to the comment of Referee 2**

We would like to thank the referee for taking the time to read our manuscript and for the thoughtful feedback. We appreciate the effort and are convinced that the reviewers' input is invaluable in enhancing the quality and rigor of our work. We acknowledge the referee's concerns regarding the discrepancies between the idealized and real data setups presented and understand the rationale behind the suggestion to split the paper into two. After careful consideration, we have decided to maintain the manuscript as a unified work as we believe that the synthetic and real-world applications are highly complementary and each would be missing the other part when published separately. But we have taken the feedback to heart and, as presented in more detail in the following sections and in the manuscript, we have re-conducted the inversions of the idealized case, aligning its setup as closely as possible with the real case inversions. We firmly believe that these modifications greatly strengthen the link between the two parts of the paper and contribute to a more coherent and concise presentation of our findings. We have taken this and the other concerns along with the comments of Referee 1 into account and present our responses below, with reviewer comments in blue and author responses in black.

The manuscript "egusphere-2023-853" presents the implementation of the model ICON-ART into the CTDAS framework. The manuscript is very well written with a very complete and thorough assessment of sensitivity.

I feel a bit uncomfortable with the presented draft. It is a very lengthy text with a lot of materials, with part of it being very technical, and the rest rather scientific with the analysis of posterior fluxes in Europe. It piles up as a text hard to follow and a bit clumsy in its structure. This is particularly regrettable as, in my opinion, it is one of the clearest and most complete presentation of the CTDAS framework that exists, with critical points, such as the propagation weights and localization lengths, being discussed and explicitly assessed.

My recommendation would be to split the manuscript into two parts: the technical description of CTDAS and sensitivity tests in OSSEs to be submitted to GMD, the application part to be kept in ACP as a much shorter paper. Such a structure would allow to further discuss sensitivity tests, better highlight the technical strengths and weaknesses of CTDAS, and become a central reference for the CTDAS community.

As explained above, we believe that maintaining a single, cohesive paper is essential for delivering a rounded understanding of our research, which includes model development, testing of the individual elements and choices in an idealized setting, and the application with real data. Two separate publications would each appear incomplete due to the missing other part.

However, we fully acknowledge the concern regarding the discrepancies between the two setups for the idealized and the real data application. We concur that these differences made it difficult to draw a straight line from the idealized (sensitivity) simulations to the real application and that this contributed to the perceived clumsiness and lack of coherence in our manuscript. In response to this concern, we re-conducted

the simulations of the synthetic study using the same setup as utilized for the real data application and tested the sensitivity to different setups starting from this reference point. We are convinced that this eliminates this weak point of the paper and greatly strengthens the connections between the two parts.

In detail, this means that we now use the same emissions (anthropogenic emissions from EDGARv6 and various natural emissions as provided by the TRANSCOM intercomparison exercise), assimilate observations at the same observation sites (and even at the exact same times), optimize the same two categories (anthropogenic and natural) and construct the model-data mismatch in the same way as in the real data application.

We further chose a new "true state" of scaling factors that is more consistent with the a priori uncertainty assumptions. Instead of scaling 11 large European regions uniformly, they are now drawn randomly (with a standard deviation of 100%) but correlated in space with a correlation length of 200 km. This approach eliminates the artificial discontinuities along country borders.

In order to save computational costs and to complete this review in a timely manner, we ran the synthetic inversions now only for 2 months instead of 7 months. This is justified as the results rapidly approach an equilibrium state with no significant further performance gain after the 2nd optimization window.

Changing the reference setup resulted in changes of the sensitivity inversions. Some inversions became obsolete, while other sensitivity-inversions arose logically. The previous table (Table 1) is therefore replaced by Table 2.

| Case | members | mdm | localization | noise | variation | bg | obs/day | window | remark |
|---|---|---|---|---|---|---|---|---|---|
| 1 | 192 | 2 ppb + 0.4*$CH_4^A$ | $\sigma$=600 km | None | None | None | 5 | 10 d | reference inversion |
| 2 | **50** | 2 ppb + 0.4*$CH_4^A$ | $\sigma$=600 km | None | None | None | 5 | 10 d | |
| 3 | 192 | **1 ppb + 0.1*$CH_4^A$** | $\sigma$=600 km | None | None | None | 5 | 10 d | |
| 4 | 192 | 2 ppb + 0.4*$CH_4^A$ | $\sigma$=600 km | **yes** | None | None | 5 | 10 d | |
| 5 | 192 | 2 ppb + 0.4*$CH_4^A$ | **None** | None | None | None | 5 | 10 d | |
| 6 | 192 | 2 ppb + 0.4*$CH_4^A$ | **$\sigma$=1000 km** | None | None | None | 5 | 10 d | |
| 7 | 192 | 2 ppb + 0.4*$CH_4^A$ | $\sigma$=600 km | None | None | None | 5 | 10 d | **uniform a priori unc.** |
| 8 | 192 | 2 ppb + 0.4*$CH_4^A$ | $\sigma$=600 km | None | None | None | 5 | 10 d | **new forecast model** |
| 9 | 192 | 2 ppb + 0.4*$CH_4^A$ | $\sigma$=600 km | None | None | None | 5 | **20 d** | |
| 10 | 192 | 2 ppb + 0.4*$CH_4^A$ | $\sigma$=600 km | None | None | **yes** | 5 | 10 d | bg uniformly scaled |
| 11 | 192 | 2 ppb + 0.4*$CH_4^A$ | $\sigma$=600 km | None | None | **yes** | 5 | 10 d | 8 $\lambda_{bg}$ |
| 12 | 192 | 2 ppb + 0.4*$CH_4^A$ | $\sigma$=600 km | None | None | **yes** | 5 | 10 d | 8 $\lambda_{bg}$(t) |
| 13 | 192 | 2 ppb + 0.4*$CH_4^A$ | $\sigma$=600 km | None | None | None | 5 | 10 d | **5 more stations** |
| 14 | 192 | 2 ppb + 0.4*$CH_4^A$ | $\sigma$=600 km | None | None | None | 1 | 10 d | **5-hr avg. conc. value** |
| 15 | 192 | **station-dependent** | $\sigma$=600 km | None | None | None | 5 | 10 d | **constant mdm at each station** |
| 16 | 192 | 2 ppb + 0.4*$CH_4^A$ | $\sigma$=600 km | None | **yes** | None | 5 | 10 d | |

Table 1

| Case | members | mdm [ppb] | localization | noise | variation | bg | window | perturbation | remark |
|---|---|---|---|---|---|---|---|---|---|
| 1 | 192 | $10 + 0.3*\overline{CH_4^{emis}}$ | $\sigma$=600 km | yes | None | None | 10 d | random | reference |
| 2 | **50** | $10 + 0.3*\overline{CH_4^{emis}}$ | $\sigma$=600 km | yes | None | None | 10 d | random | |
| 3 | 192 | $10 + 0.3*\overline{CH_4^{emis}}$ | $\sigma$=600 km | **None** | None | None | 10 d | random | |
| 4 | 192 | $10 + 0.3*\overline{CH_4^{emis}}$ | **None** | yes | None | None | 10 d | random | |
| 5 | 192 | $10 + 0.3*\overline{CH_4^{emis}}$ | **$\sigma$=1200 km** | yes | None | None | 10 d | random | |
| 6 | 192 | $10 + 0.3*\overline{CH_4^{emis}}$ | $\sigma$=600 km | yes | None | None | 10 d | **11 regions** | |
| 7 | 192 | $10 + 0.3*\overline{CH_4^{emis}}$ | $\sigma$=600 km | yes | None | None | 10 d | random | **full state propagation** |
| 8 | 192 | $10 + 0.3*\overline{CH_4^{emis}}$ | $\sigma$=600 km | yes | None | None | **20 d** | random | |
| 9 | 192 | $10 + 0.3*\overline{CH_4^{emis}}$ | $\sigma$=600 km | yes | None | **yes** | 10 d | random | bg uniformly scaled |
| 10 | 192 | $10 + 0.3*\overline{CH_4^{emis}}$ | $\sigma$=600 km | yes | None | **yes** | 10 d | random | $8\,\lambda_{bg}$ |
| 11 | 192 | $10 + 0.3*\overline{CH_4^{emis}}$ | $\sigma$=600 km | yes | None | **yes** | 10 d | random | $8\,\lambda_{bg}(t)$ |
| 12 | 192 | $10 + 0.3*\overline{CH_4^{emis}}$ | $\sigma$=600 km | yes | None | None | 10 d | random | **5 more stations** |
| 13 | 192 | **$10 + 0.3*CH_4^{emis}$** | $\sigma$=600 km | yes | None | None | 10 d | random | |
| 14 | 192 | $10 + 0.3*\overline{CH_4^{emis}}$ | $\sigma$=600 km | yes | **yes** | None | 10 d | random | |
| 15 | 192 | $10 + 0.3*\overline{CH_4^{emis}}$ | $\sigma$=600 km | yes | None | None | 10 d | random | **3 emission categories** |

Table 2

Please find below general recommendations and technical remarks on the present manuscript.

1. **Points to elaborate on**

- p8, l201: Previous assimilation steps are propagated towards further assimilation steps with a weighting factors of 1/3 (or 2/3). The only justification for this weighting points to van der Laan-Luijkx et al. (2017). To my knowledge, this paper only mentions the weights in their equation (3), with no specific justification. The weights are practical tweaks to smooth posterior fluxes from one period to the other. However, to my knowledge, they are never rigorously justified. This is a critical over-sight by the overall CTDAS community, but still needs to be address. The present paper is arguably one of the most clearly written one from the CTDAS heritage, but shortcomings still need to be addressed. Please justify further these weights factors. Moreover, as there is no clear justification for the weights altogether, why using different weights between the real data application case and the pseudo-data application? This point is partially addressed in the sensitivity tests, but should be further highlighted

The referee is right that this is a weak point that has not received sufficient attention in previous publications and in ours. Obviously, the larger the value the more strongly the previous step is propagated forward. The choice of 2/3 in van der Laan-Luijkx (2017) is based on the concept that the system should revert to the prior if no data is available to constrain it, whereas the timescale to revert was a design choice. By choosing 2/3, a lambda of 2 would fall below 1.1 within 8 weeks, which seems reasonable.
We have adopted this relatively high value of 2/3 as we believe that there is valuable information on CH4 emissions in the previous optimization step that should be propagated forward, in particular

because CH4 emissions from key sources such as agriculture are not expected to change drastically from one assimilation window to the next. A value of 2/3 is a pragmatic balance between propagation of information and constraining the emissions in a given time window primarily by observations in the same (and next) window. When propagating the state in this way, also the uncertainties should be propagated. However, doing this in a way that avoids an unrealistic contraction of uncertainties is challenging. It would have to include an inflation of uncertainties due to the errors in the assumption of temporal persistence of emissions as proposed e.g. by Brunner et al. (2012). Not propagating uncertainties but rather revert to the a priori uncertainties was again a pragmatic and safe choice. It avoids the issue of unrealistic contraction, but it tends to overestimate the a priori uncertainties especially in regions where previous steps provided strong constraints. It could indeed be useful to apply a more sophisticated approach, but we feel that this requires a much more comprehensive investigation that would have to address challenging questions such as temporal variability of CH4 sources. In a new CTDAS version, it is possible to use temporal covariations. The ensemble members are not drawn from P as white noise, but rather as red noise with an autocorrelation length.

We have added one sentence in the manuscript that shortly explains the motivation for propagating the posterior with 2/3.

- 2.2.7: The localization is, with the propagation weights, one of the big practical tweaks needed for CTDAS to work properly. Such method was introduced in Houtekamer and Mitchell (1998) to avoid numerical artifacts due to the small number of Monte Carlo members compared to the degrees of freedom in the problem. In Houtekamer and Mitchell (1998), they provide an assessment of the best cutoff term. Such a value (here 600km) could greatly influence the final results of the inversion. The sensitivity to the localization is tested in the sensitivity tests but deserves stronger highlight in the text.

We tested the localization length with two sensitivity inversions, where we tested once with twice the length (i.e. 1200 km) and once without localization at all. In the inversion without localization, we see that the performance deteriorates significantly based on the 10-day optimization window. We conclude that without localization, a significant amount of noise in the Kalman gain matrix actually degrades our results. The performance of the optimization based on the values averaged over the entire period, on the other hand, hardly differs from the reference inversion, from which we conclude that this noise is really random and disappears when averaged over multiple optimization windows.

Doubling the localization length has little impact on the category with large emission fluxes, but slightly worsens the optimization of natural emissions (with smaller fluxes). This is to be expected because with a larger localization length we take a larger area into account and thus allow more noise in the Kalman gain matrix. Since this noise has more weight compared to the signal in categories with small fluxes, we see a larger effect of the large localization length in the natural emissions.

From our sensitivity inversions we conclude that the localization length of 600 km is a good choice.

In the revised manuscript we discuss these findings in more detail as suggested by the reviewer.

**1. Technical comments**

- Please improve consistency in naming your data assimilation method. The following terms are used at different parts of the text: Ensemble Kalman Filter, Ensemble Square-Root filter and Ensemble Kalman Smoother. The atmospheric community often use the three terms indiscriminately and incorrectly, which should be fixed at some point... From my point of view (with no certainty from my side), CTDAS is incorrectly called EnKF in the community, whereas it is based on an Ensemble Square-Root Filter, as originally specified in Peters et al., 2005 and therein referencing Whitaker and Hamill (2002) (https://doi.org/10.1175/1520-0493(2002)130<1913:EDAWPO>2.0.CO;2). Please clarify this point, justify the naming and stick to it all over the text.

In fact, the correct name is "ensemble square root filter" (EnSRF) but also "ensemble Kalman smoother" (EnKS) would be correct as the EnSRF is just a special form of an EnKS. We thank the referee for pointing this out. In accordance with van der Laan-Luijkx (2017), we now use the term "ensemble Kalman smoother" or "EnKS" throughout our manuscript and abandon the terms "ensemble Kalman filter" and "EnKF".

- The Online Emissions Module seems to be a powerful tool; however, it seems that temporal and vertical factors are uniformly applied over the grid. Does this mean that all the pixels have the same temporal patterns? How such a module would deal with complex emission patterns, such as wild fires? Is there at least different temporal patterns over different regions of the domain of interest? e.g., by country or finer?

That's correct, for our application with methane we apply constant temporal profiles and use a vertical profile in which emissions only occur in the lowest 20 m. In this respect, we do not use all the possibilities that OEM would offer. OEM allows prescribing distinct hourly, daily or monthly periodic profiles (or alternatively hourly profiles for the whole year) per emission category and per country or any other regions defined by masks. For CO2, we also implemented the Vegetation Photosynthesis and Respiration Model (VPRM) to enable inverse estimation of these not strictly periodically varying fluxes.

For complex emissions such as wild fires, however, OEM has its limitations. In that case the perturbations are best applied to externally generated emission input files. ICON-ART can flexibly combine emissions treated by OEM with emissions read from external input files.

- What is the overall computation cost of the system? This is an important component to communicate to put it in perspective to the quality of the results.

  The computational costs are very much dependent on the architecture of the computer, the version of the ICON model and the setup and configuration of the simulations. We carried out the simulations on the supercomputer "Piz Daint" of the Swiss National Supercomputing Center (CSCS), namely on the XC40 compute nodes, each with two Intel Xeon E5-2695 v4 2.10GHz processors (2x18 cores, 64/128 GB RAM). Typically we ran the simulations spread across 16 nodes. A simulation with the described setup and a resolution of approx. 26 km required around 10 node hours per 11-day simulation. The computational costs for CTDAS are comparatively low with around 5 node hours per assimilation cycle (running on one node). For the inversion of a whole year, the total costs amounted to around 1300 node hours or 36 x 1300 = 46'800 core hours. We expect these numbers to strongly drop (possibly by a factor 10) in the next year for two reasons: First, we have developed a new GPU-version of ICON-ART leveraging previous developments of a GPU-version of ICON (Giorgetta et al. 2022) and second, a new supercomputer Alps will replace Piz Daint in 2024 with a completely new hardware architecture offering dramatic performance gains for GPU applications (https://www.cscs.ch/science/computer-science-hpc/2021/cscs-hewlett-packard-enterprise-and-nvidia-announce-worlds-most-powerful-ai-capable-supercomputer). We will briefly mention the computational cost of the current combination of model system and hardware in the revised manuscript.

- The use of a constant offset to deal with negative values in ICON-ART generated by the ensemble is based on the assumption that ICON-ART is purely linear. Passive transport is theoretically linear, but numerical models often introduce non-linearity; has it been tested? what is the impact on the simulations?

  Indeed, the transport of passive tracers in ICON-ART is not completely linear. However, the impact of adding a constant offset of 1.2e-6 is very small. The 95th percentile of the absolute differences (in mass mixing ratios) is ca. 1.0e-12, while the 99th percentile reaches 3.4e-11. In term of volume mixing ratios, this corresponds to ca. 0.06 ppb. We consider these differences to be insignificantly small and we don't expect substantial impact on the outcomes of our inversions. We now mention this in a sentence in the manuscript.

- p7, l 206: Emissions are assumed to influence observed concentration for a maximum of 20 days in Europe. It is most likely true and should not be presented as an assumption; a very simple sensitivity test could be done and confirm once for all the assumption.

  We carried out the following test: In a simulation over 31 days with a methane tracer each for anthropogenic and natural emissions, we ran the first 11 days with the standard EDGAR and natural

emissions but then stopped the emissions and looked at how much mass of the emitted tracer remains in the domain over the next 20 days. The total mass relative to the end of day 11 is plotted in Figure 1. After 10 days only 0.26% of the tracer mass is still remaining, after 20 days it is only 0.001%, which means that there is absolutely no significant signal of the emissions remaining in the domain after 20 days. We re-formulated the text and present it as a fact instead of an assumption. Although this result will vary depending on weather situation, we know from extensive studies with the Lagrangian transport model FLEXPART-COSMO (e.g. Bergamaschi et al., 2022) that in most situations very little tracer mass remains over Europe after 8 days of simulation, consistent with the present experiment.

[Figure]

Figure 1

- A schematic of the observation operator, with perturbation, inputs, scaling factors, etc., and interactions between CTDAS, ICON and OEM would be needed to fully clarify the data flow in your inversion system.

We have added a corresponding schematic (Fig. 2). It shows the required inputs for the ICON-ART simulations and the inputs for CTDAS. It also shows the interaction between ICON-ART and CTDAS, i.e. that CTDAS creates the perturbation factors for the flux ensemble in ICON-ART and how the sampled concentrations of the a priori emissions and the flux ensemble acts as input for CTDAS. The schematics of the operating principle of the ensemble Kalman smoother is already in the manuscript (Fig. 1).

[Figure]

Figure 2

- p12 l.308: It is stated that no temporal correlation is applied between assimilation windows; However the weights applied to prior fluxes using posterior fluxes from previous assimilation steps have the same impact as temporal correlations. This should be clarified and acknowledged.

This is true. What we wanted to express is that we assume no temporal correlations in the covariance and start with the new a priori covariance matrix in each 10-day window. We have adapted the text in the manuscript accordingly to express this more clearly.

- 2.2.5 Idealized case: scaling emission uncertainties to fit country-scale relative uncertainties is, in my opinion, a mis-interpretation of what TNO does in their inventory; I agree with the rational, but I am skeptical on the conclusion that, e.g., emissions in the Netherlands are much more uncertain than emissions in larger countries, e.g., Poland. There is an ever-lasting debate on what covariance structures we should use in atmospheric inversions and there is no proper definite answer. Still, this should be further discussed and it might be in the end more relevant to do the inversion with scaling factors at the country scale to fit the uncertainties as provided by TNO; otherwise, there is no possible equivalent between what TNO gives as uncertainties and what is prescribed in the inversion.

The importance of realistic covariance structures in a priori emissions is receiving increasing attention in the inverse modelling community as shown by the recent publications by Super et al. (2020) and Choulga et al. (2021). Because of potentially different choices of emission factors in different

countries (IPCC default values or country-specific Tier-2 factors), emission uncertainties in the TNO inventory can indeed be correlated at the country scale. However, this assumption does not hold for EDGAR, which applies the same methods across countries.

As we have changed the emissions in the idealized case to the same emissions as in the real case (EDGAR plus various natural sources), we accordingly changed the covariance structure. This discussion has therefore become obsolete in our publication because we no longer use the TNO inventory and the covariance structure mentioned.

- In the end, my biggest worry is not about the values themselves that could be infinitely argued on, but rather on the strong inconsistencies between conclusions on the academic case and the real-data case. Why taking different prior inventory with so much different error structures? It makes interpretation difficult.

As mentioned above, we have carried out a major revision, adapting the setup of the synthetic studies to the real setup.

- p12 l322: isn't it the posterior that is propagated with 100% weight rather than the prior?

That is correct. We fixed that.

**References:**

Brunner, D., Henne, S., Keller, C. A., Reimann, S., Vollmer, M. K., O'Doherty, S., and Maione, M.: An extended Kalman-filter for regional scale inverse emission estimation, Atmos. Chem. Phys., 12, 3455–3478, https://doi.org/10.5194/acp-12-3455-2012, 2012.

Choulga, M., Janssens-Maenhout, G., Super, I., Solazzo, E., Agusti-Panareda, A., Balsamo, G., Bousserez, N., Crippa, M., Denier van der Gon, H., Engelen, R., Guizzardi, D., Kuenen, J., McNorton, J., Oreggioni, G., and Visschedijk, A.: Global anthropogenic $CO_2$ emissions and uncertainties as a prior for Earth system modelling and data assimilation, Earth Syst. Sci. Data, 13, 5311–5335, https://doi.org/10.5194/essd-13-5311-2021, 2021.

Giorgetta, M. A., Sawyer, W., Lapillonne, X., Adamidis, P., Alexeev, D., Clément, V., Dietlicher, R., Engels, J. F., Esch, M., Franke, H., Frauen, C., Hannah, W. M., Hillman, B. R., Kornblueh, L., Marti, P., Norman, M. R., Pincus, R., Rast, S., Reinert, D., Schnur, R., Schulzweida, U., and Stevens, B.: The ICON-A model for direct QBO simulations on GPUs (version icon-cscs:baf28a514), Geosci. Model Dev., 15, 6985–7016, https://doi.org/10.5194/gmd-15-6985-2022, 2022.

Super, I., Dellaert, S. N. C., Visschedijk, A. J. H., and Denier van der Gon, H. A. C.: Uncertainty analysis of a European high-resolution emission inventory of $CO_2$ and CO to support inverse modelling and network design, Atmos. Chem. Phys., 20, 1795–1816, https://doi.org/10.5194/acp-20-1795-2020, 2020.

van der Laan-Luijkx, I. T., van der Velde, I. R., van der Veen, E., Tsuruta, A., Stanislawska, K., Babenhauserheide, A., Zhang, H. F., Liu, Y., He, W., Chen, H., Masarie, K. A., Krol, M. C., and Peters, W.: The CarbonTracker Data Assimilation Shell (CTDAS) v1.0: implementation and global carbon balance 2001–2015, Geosci. Model Dev., 10, 2785–2800, https://doi.org/10.5194/gmd-10-2785-2017, 2017.

---

## Referee Report (RR1)

**European CH4 inversions with ICON-ART coupled to CarbonTracker Data Assimilation Shell**

Michael Steiner et al.

The manuscript is a revision of a paper describing experiments with inverse modelling of methane emissions over Europe using the ICON-ART model.

In the new version, the focus is more on the final application with real data, and less on the initial experiments with synthetic data. This makes the practical application more clear and is therefore certainly of added value.

A major change is that where the previous described experiments distinguish 3 different source categories (agriculture, waste, and remaining sources), the new experiments distinguish only 2 (anthropogenic and natural). As these two are spatially more disjunct than the previously distinguished sources, the results are more robust. The results give confidence that the described system is able to reduce at least part of the uncertainty in the total anthropogenic emissions. On the other hand, it is also illustrated that it is not yet possible yet to reduce the uncertainty in the natural emissions too.

Many minor corrections were made to the manuscript, for example in consequent description of the inversion system as a Kalman *smoother* rather than a *filter*.